

# Online model calibration for a simplified LES model in pursuit of real-time closed-loop wind farm control

Bart Doekemeijer[1], Sjoerd Boersma[1], Lucy Pao[2], Torben Knudsen[3], and Jan-Willem van Wingerden[1]

[1]Delft Center for Systems and Control, Delft University of Technology, The Netherlands
[2]Electrical, Computer & Energy Engineering, University of Colorado Boulder, Colorado, United States of America
[3]Department of Electronic Systems, Aalborg University, Denmark

**Correspondence:** B.M. Doekemeijer (B.M.Doekemeijer@tudelft.nl)

**Abstract.** Wind farm control often relies on computationally inexpensive surrogate models to predict the dynamics inside a farm. However, the reliability of these models over the spectrum of wind farm operation remains questionable due to the many uncertainties in the atmospheric conditions and tough-to-model flow dynamics at a range of spatial and temporal scales relevant for control. A closed-loop control framework is proposed in which a simplified dynamical LES model is calibrated and used for optimization in real time. This paper presents an estimation solution with an Ensemble Kalman filter (EnKF) at its core, which calibrates the surrogate model to the actual atmospheric conditions. The estimator is tested in high-fidelity simulations of a nine-turbine wind farm. Using exclusively turbine SCADA measurements, the adaptability to modeling errors and changes in atmospheric conditions (TI, wind speed) is shown. Convergence is reached within $400$ seconds of operation, after which the estimation error in flow fields is negligible. At a low computational cost of $1.2$ s on an $8$-core CPU, this algorithm shows comparable accuracy to the state of the art from the literature while being approximately two orders of magnitude faster. Using the calibration solution presented, the surrogate model can be used for accurate forecasting and optimization.

## 1 Introduction

Over the past decades, global awakening on climate change and the environmental, political and financial issues concerning fossil fuels have been catalysts for the growth of the renewable energy industry. As the primary energy demand in Europe is projected to decrease by 200 million tonnes of oil equivalent from 2016 to 2040, there is an additional shift in the energy source used to meet this demand (International Energy Agency, 2017). Shortly after 2030, onshore and offshore wind energy are projected to become the main source of electricity for the European Union. By then, about $80\%$ of all new capacity added is projected to come from renewable energy sources, enabled by a favorable political climate.

While there are clear benefits in the growth of the wind energy industry, an important problem with wind energy is that almost all commercial turbines are currently disconnected from the electricity grid by their power electronics (Aho et al., 2012). As the current grid-connected fossil fuel plants are replaced by grid-disconnected renewable energy plants, the inertia of the electricity grid will decrease. Thus, the grid will become less stable, making it more prone to machine damage and blackouts (Ela et al., 2014). Therefore, there is a strong need for wind farms and other renewables to provide ancillary grid services. Wind farm control aimed at increasing the grid stability is more commonly defined as active power control (APC). In



APC, the power production of a wind farm is regulated to meet the power demand of the electricity grid, which may change from second to second.

Existing literature on wind farm control has focused mainly on maximizing the power capture (e.g., Rotea, 2014; Gebraad and van Wingerden, 2015; Gebraad et al., 2016; Annoni et al., 2016a; Munters and Meyers, 2017; Vali et al., 2017). Though,
literature on APC has been receiving an increasing amount of attention (e.g., Fleming et al., 2016; Van Wingerden et al., 2017; Boersma et al., 2017a). The main challenges in wind farm control are the large time delays caused by the formation of wakes, the many uncertainties in the atmospheric conditions, and the questionable reliability of surrogate models over the wide spectrum of wind farm operation (see Boersma et al. (2017a) and Knudsen et al. (2015) for state-of-the-art overviews of control and control-oriented modeling for wind farms). While there has been success with model-free methods for power maximization
(e.g., Rotea, 2014), it is unclear to what degree such methods can be used for power forecasting. Furthermore, model-free methods typically have long settling times, making these methods intractable for APC. On the other hand, for model-based approaches, the aforementioned challenges make it impossible for any model to reliably provide power predictions in an open-loop setting. Hence, a model-based approach in which a surrogate wind farm model is actively adjusted to the present conditions is a necessity for reliable and computationally tractable APC algorithms. This closed-loop wind farm control framework is
displayed in Fig. 1. The control framework of Fig. 1 requires three components.

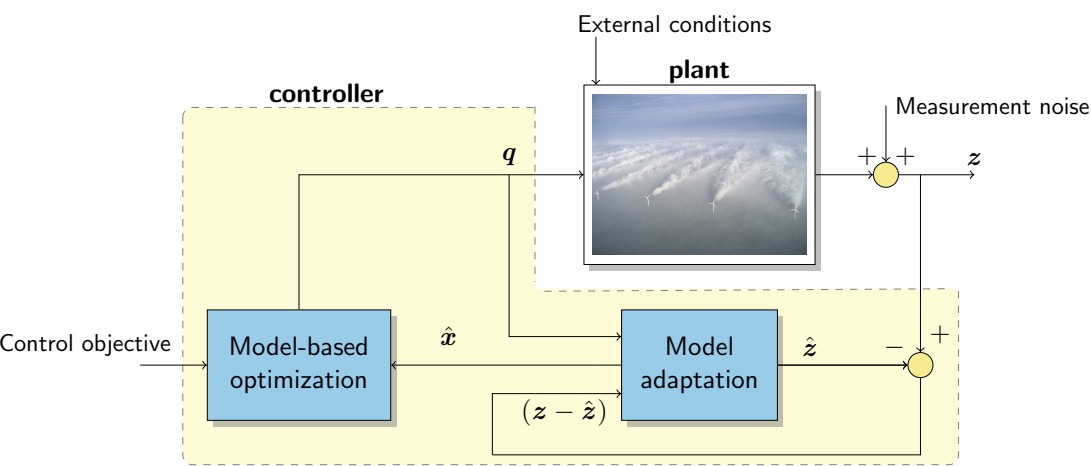

**Figure 1.** Closed-loop wind farm control framework. Measurements $z$ (e.g., SCADA, met mast, LiDAR data) are fed into the controller block. First, the state of the surrogate wind farm model $x$ is estimated to represent the actual atmospheric and turbine conditions inside the wind farm. Secondly, using the calibrated model, an optimization algorithm determines the control policy (e.g., yaw angles, blade pitch angles) for all turbines $q$. This control policy may be a set of constant operating points, but can also be time-varying, depending on whether the surrogate model is time-varying and the employed optimization algorithm.

The first component of the closed-loop framework is a computationally inexpensive control-oriented surrogate model that accurately predicts the power production of the wind farm ahead in time, on a time-scale relevant for control. The most





commonly used surrogate models in wind farm control are steady-state models, which are heuristic and neglect all temporal dynamics (Boersma et al., 2017a). Thus, wind farm control algorithms synthesized using such models neglect any transient dynamics in the wind farm, thereby potentially limiting performance. While some of these models have shown success in wind tunnel tests (e.g., Schreiber et al., 2017) and field tests (e.g., Fleming et al., 2017a, b) for power maximization, the actuation

frequency is limited to the minutes-scale, since the flow and turbine dynamics are predicted on the minute-scale. Furthermore, time-ahead predictions with these models are limited to the time-invariant steady-state, limiting their use for APC.[1] There is a smaller yet significant number of dynamic surrogate wind farm models (e.g., Munters and Meyers, 2017; Boersma et al., 2017b; Shapiro et al., 2017a), which attempt to model the dominant temporal dynamics inside the farm. These models can be used for control on the seconds-scale, and furthermore allow time-ahead predictions, some even under changing atmospheric conditions.

Specifically, the dynamic surrogate model employed in Shapiro et al. (2017a) is computationally feasible, but only models the flow in one dimension, and furthermore allows no turbine yaw or changes in the wind direction, limiting its applicability. Furthermore, the dynamical model in Munters and Meyers (2017) has shown success for closed-loop control applications, but as it is a 3D LES code, it is much too computationally costly for any kind of real-time control, and the authors present their results solely as a benchmark case. In the work presented here, the model described in Boersma et al. (2017b) is used, which is

a light-weight two-dimensional LES code with wind farm control as its main objective. This dynamic surrogate model, named "WindFarmSimulator" (WFSim), includes yaw and axial induction actuation, turbine-induced turbulence effects, and spatially and temporally varying inflow profiles, with a moderate computational cost ($10^1 - 10^2$ ms per timestep).[2]

The second component of the closed-loop framework is an algorithm that adjusts the surrogate model's parameters to improve its accuracy online using flow and/or turbine measurements (e.g., SCADA data, LiDAR measurements, met masts). In

terms of control, this turns into a dual estimation problem, in which both the model state and a subset of model parameters are estimated online. Currently, the optimization algorithms presented in Munters and Meyers (2017) and Vali et al. (2017) have assumed full state knowledge, conveniently ignoring the step of model adaptation. Literature on state reconstruction and model calibration for dynamical wind farm models is sparse, limited to linear low-order models and/or common estimation algorithms. Gebraad et al. (2015) designed a traditional Kalman filter (KF) for their low-fidelity "FLORIDyn" model, sho-

wing marginal improvements compared to optimization using a static model. Shapiro et al. (2017a) present a one-dimensional dynamic wake model used with receding horizon control for secondary frequency regulation, using an estimation algorithm following Doekemeijer et al. (2016). Furthermore, Iungo et al. (2015) used dynamic mode decomposition to obtain a reduced-order model of the wind farm dynamics, which was then combined with a traditional KF for state estimation. To the best of the authors' knowledge, none of these methods have explored more sophisticated models such as WFSim, and often only use

simple state estimation algorithms that are lacking in terms of accuracy and in terms of computational tractability.

The third component of the closed-loop framework is an optimization algorithm, which typically is a gradient-based or nonlinear optimization algorithm (e.g., Gebraad et al., 2016; Thomas et al., 2017) for steady-state models, and a model-based

---

[1]Control using steady-state models is typically limited to an actuation frequency of every 5-10 minutes, depending on the wind speed and size of the farm. Namely, after each change in control settings, it takes minutes before the flow propagates through the farm and a steady-state is formed. In steady-state, SCADA data is then temporally averaged, upon which a new set of control parameters is calculated for the current atmospheric conditions.

[2]For a more detailed analysis on the computational cost involved in WFSim, the reader is referred to Boersma et al. (2017b).



predictive optimization method for dynamical models (e.g., Goit and Meyers, 2015; Vali et al., 2017; Siniscalchi-Minna et al., 2018). A more in-depth discussion on optimization algorithms for the framework of Fig. 1 is out of the scope of this article, and therefore not further continued here.

The focus of this work is on a model adaptation algorithm for WFSim, which trades off estimation accuracy with computa-
tional complexity for online model calibration. In previous work (Doekemeijer et al., 2016, 2017), recursive state estimation using flow measurements downstream of each turbine has shown success using an Ensemble KF (EnKF), with a computational cost several orders of magnitude lower than traditional KF methods. The main contributions of this article are 1) the addition of adaptation to time-varying atmospheric conditions (specifically, the freestream wind speed and turbulence intensity), which is of crucial importance for accurate longer-term forecasting, 2) each turbine's power signal can now be used in addition to, or
instead of, flow measurements, as power measurements are readily available in existing farms, in contrast to flow measurements (other than the hub anemometer, which yields low-quality measurements), 3) the computational complexity is further reduced compared to previous work, and 4) the EnKF algorithm will be compared to the state of the art algorithms in the literature.

The structure of this article is as follows. In Section 2, the surrogate model will be described in more detail. In Section 3, a time-efficient, online model calibration algorithm for low- and medium-fidelity dynamical wind farm models is detailed.
This online calibration algorithm is tested against standard algorithms in the literature using high-fidelity simulation data in Section 4. The article is concluded in Section 5.

## 2    The surrogate model

To motivate the choice of model parameters that are to be estimated in real time, the surrogate model used in the work at hand is outlined in this section. The chosen surrogate model is the WindFarmSimulator (WFSim) model presented by Boersma et al.
(2017b). In short, WFSim solves a modified set of unsteady two-dimensional (2D) Navier-Stokes equations in a horizontal plane at the turbine hub height. This surrogate model is a medium-fidelity nonlinear dynamic wind farm model with a total of 5 tuning parameters. WFSim has shown success in reconstructing the flow field and turbine power signals of high-fidelity LES data. This model is particularly suited for the framework presented in Fig. 1 as it is dynamic, includes both yaw and axial induction control, handles temporally and spatially varying inflows, and yields a relatively high accuracy with a manageable
computational cost.

In Section 2.1, the governing equations of the model are presented. The turbulence and turbine model are described in Sections 2.2 and 2.3, respectively. The spatial and temporal discretization process is described in Section 2.4, including some remarks about the boundary conditions and the computational tractability of the model.

### 2.1   Governing equations

The WFSim wind farm model is based on the two-dimensional unsteady incompressible Navier-Stokes (NS) equations to maintain computational tractability compared to a three-dimensional model. Furthermore, in WFSim the continuity equation is modified to accommodate for flow dissipation in the neglected vertical dimension. The surrogate model can be described





completely by the flow and rotor dynamics in a horizontal plane at hub height, derived from the following set of partial differential equations:

$$\frac{\partial \boldsymbol{u}}{\partial t} + (\boldsymbol{u} \cdot \boldsymbol{\nabla}_H)\boldsymbol{u} + \boldsymbol{\nabla}_H \cdot \boldsymbol{\tau}_H + \boldsymbol{\nabla}_H \cdot p = \mathbf{f}, \qquad \frac{\partial u}{\partial x} + 2\frac{\partial v}{\partial y} = 0, \qquad \text{where} \quad \boldsymbol{u} = \begin{bmatrix} u & v \end{bmatrix}^T, \quad \boldsymbol{\nabla}_H = \begin{bmatrix} \frac{\partial}{\partial x} & \frac{\partial}{\partial y} \end{bmatrix}^T, \qquad (1)$$

where $u$ and $v$ are the longitudinal and lateral flow velocity respectively, $x$ and $y$ are the spatial coordinates in longitudinal and lateral direction respectively, $\boldsymbol{\tau}_H$ is a 2D tensor containing the horizontal subgrid stresses (turbulence model), $p$ is pressure, and $\mathbf{f}$ contains the forcing terms (turbine model) acting on the flow. Equation (1) deviates from the traditional 2D NS equations in two ways. Firstly, the diffusion term is neglected, as it plays a negligible role in the dominant flow dynamics due to the low viscosity of air. Secondly, the term $\frac{\partial v}{\partial y}$ in the continuity equation is multiplied by a factor 2 to approximate flow dissipating in the vertical flow dimension. See Boersma et al. (2017b) for a detailed derivation of (1).

## 2.2 Subgrid-scale (turbulence) model

Boersma et al. (2017b) introduced a new model for the subgrid-scale term $\boldsymbol{\tau}_H$. The subgrid-scale model is formulated using an eddy-viscosity assumption in combination with Prandtl's mixing length model,

$$\boldsymbol{\tau}_H = -\ell_u(x,y)^2 \left| \frac{\partial u}{\partial y} \right| \cdot \left( \frac{1}{2}\boldsymbol{\nabla}_H \boldsymbol{u} + (\boldsymbol{\nabla}_H \boldsymbol{u})^T \right), \quad \text{with} \quad \ell_u(x,y) = \begin{cases} G(x_i', y_i') * \ell_u^i(x_i', y_i'), & \text{if } x \in \mathcal{X} \text{ and } y \in \mathcal{Y}, \\ 0, & \text{otherwise}, \end{cases} \qquad (2)$$

where $\ell_u(x,y) \in \mathbb{R}^+$ is a local spatially varying parametrization of the mixing length, inspired by the high-fidelity simulation results presented in Iungo et al. (2017). $G(x_i', y_i')$ is a smoothing pillbox filter with radius 3, $*$ is the 2D spatial convolution operator, and $\mathcal{X}$ and $\mathcal{Y}$ define a rectangular region behind the turbine rotor to which the turbulence model applies, given by

$$\mathcal{X} = \{x : x_i' \leq x \leq x_i' + \cos(\phi) \cdot d\}, \quad \mathcal{Y} = \{y : y_i' - \frac{D}{2} + \sin(\phi) \cdot x_i' \leq y \leq y_i' + \frac{D}{2} + \sin(\phi) \cdot x_i'\},$$

with $(x_i', y_i')$ the wind-aligned axis system centered at the turbine rotor, $D$ the turbine rotor diameter, $\phi$ the mean wind direction in the original $(x,y)$-axis system, and $d$ a length parameter for the turbulence model. See Fig. 2 for a schematic drawing. Then, $\ell_u^i(x,y)$ is defined as

$$\ell_u^i(x_i', y_i') = \begin{cases} (x_i' - d')\ell_s, & \text{if } d' \leq x_i' \leq d \text{ and } -\frac{D}{2} \leq y_i' \leq \frac{D}{2}, \\ 0, & \text{otherwise}, \end{cases} \qquad (3)$$

where $\ell_s$ defines the slope of $\ell_u^i(x_i', y_i')$, and $d'$ is a second length parameter for the turbulence model. Thus, the entire turbulence model has three tuning parameters: the length parameters $d$ and $d'$ are the upper and lower spatial bounds, respectively, and $\ell_s$ is a gradient parameter for the mixing length.





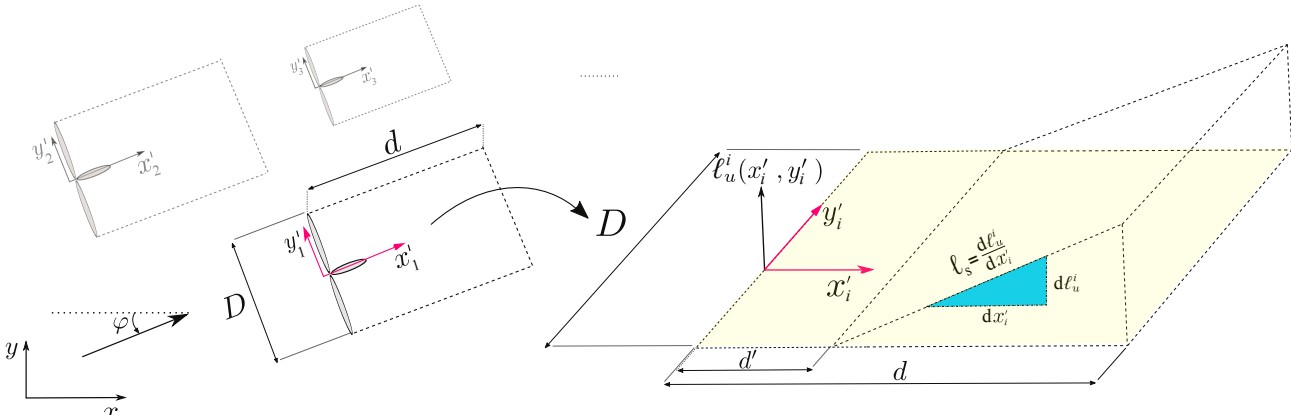

**Figure 2.** The subgrid-scale model implemented in WFSim employs a spatially varying mixing length parameter that increases with distance behind the rotor. This can be explained by the turbine-induced turbulent structures in the wake. Image courtesy of Boersma et al. (2017b).

## 2.3 Turbine model

Turbine forces in WFSim are modeled using the classical non-rotating actuator disk model (ADM), projected onto the 2D plane at hub height. The turbine forcing term in (1), $\mathbf{f}$, at spatial location $\boldsymbol{s} = \begin{bmatrix} x & y \end{bmatrix}^T \in \mathbb{R}^2$ is expressed as

$$\mathbf{f} = \sum_{i=1}^{N_T} \mathbf{f}_i, \quad \text{with} \quad \mathbf{f}_i = \frac{c_f}{2} C'_{T_i} \left[ U_i \cos(\gamma_i) \right]^2 \begin{bmatrix} \cos(\gamma_i + \phi) \\ \sin(\gamma_i + \phi) \end{bmatrix} \cdot \mathrm{H} \left[ \frac{D}{2} - ||\boldsymbol{s} - \boldsymbol{t}_i||_2 \right] \cdot \delta \left[ (\boldsymbol{s} - \boldsymbol{t}_i) \cdot \boldsymbol{e}_{\perp,i} \right], \quad (4)$$

with H[•] the heaviside function, $\delta[\bullet]$ the Dirac delta function, and $\boldsymbol{e}_{\perp,i} \in \mathbb{R}^2$ the unit vector perpendicular to the $i^{\text{th}}$ rotor disk with position $\boldsymbol{t}_i \in \mathbb{R}^2$. The scalar $C'_{T_i}$ is a variation of the non-dimensional thrust coefficient of turbine $i$ which can be related to physical turbine parameters such as the generator torque and blade pitch angles (Goit and Meyers, 2015). The scalar $\gamma_i$ is the yaw angle of turbine $i$ with respect to the incoming wind, and $U_i$ is the average flow velocity over the rotor of turbine $i$. The scalar $c_f$ is a static tuning variable to account for the time-invariant rotor dimensions and numerical grid effects, making

it the fourth tuning variable in WFSim. The control variables for optimization in the framework of Fig. 1 are $\gamma_i$ and $C'_{T_i}$ for $i = 1, ..., N_T$, with $N_T$ the number of turbines. Furthermore, the instantaneous power capture of the wind farm $P_{\text{farm}}$ is calculated in a similar approach by

$$P_{\text{farm}} = \sum_{i=1}^{N_T} P_{\text{turb},i}, \quad \text{with} \quad P_{\text{turb},i} = \frac{c_p}{2} \rho A C'_{T_i} \left[ U_i \cos(\gamma_i) \right]^3, \quad (5)$$

with scalar $c_p$ the fifth tuning factor used to account for numerical grid effects and time-invariant turbine losses, and $A$ the rotor

swept surface area. Note that $C'_T$ has a direct mapping to the turbine power $P_{\text{turb},i}$, and thus replaces the usual non-dimensional power coefficient, following the example of Goit and Meyers (2015).



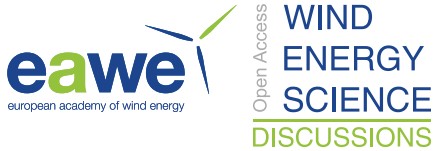

### 2.4 Discretization, boundary conditions, and computational cost

Equation (1) is spatially discretized on a quadrilateral grid employing the finite volume method and the hybrid differencing scheme (Boersma et al., 2017b). Temporal discretization is performed using the implicit method, which guarantees stability of the solution. Dirichlet boundary conditions for $u$ and $v$ are applied on one side of the grid for inflow, while Neumann boundary conditions are applied on the remaining sides for the outflow. After discretization, the surrogate wind farm model described in this section reduces to a nonlinear discrete-time deterministic state-space model, described by

$$\boldsymbol{x}_{k+1} = f(\boldsymbol{x}_k, \boldsymbol{q}_k),$$
$$\boldsymbol{z}_k = h(\boldsymbol{x}_k, \boldsymbol{q}_k),$$

where $\boldsymbol{x}_k \in \mathbb{R}^N$ is the system state at discrete time instant $k \in \mathbb{Z}$, which is a column vector containing the collocated longitudinal flow velocity at each cell in the domain $\boldsymbol{u}_k \in \mathbb{R}^{N_u}$, the lateral flow velocity at each cell in the domain $\boldsymbol{v}_k \in \mathbb{R}^{N_v}$, and the pressure term at each cell in the domain $\boldsymbol{p}_k \in \mathbb{R}^{N_p}$, with $N = N_u + N_v + N_p$ and $N_u \approx N_v \approx N_p \approx \frac{1}{3}N$. The state $\boldsymbol{x}_k$ is formulated as

$$\boldsymbol{x}_k^T = \begin{bmatrix} \boldsymbol{u}_k^T & \boldsymbol{v}_k^T & \boldsymbol{p}_k^T \end{bmatrix}.$$

Empirically, good results have been achieved with cell dimensions of about $30-50$ m in width and length, resulting in $N$ with a typical value on the order of $10^3 - 10^4$ for medium-sized wind farms (e.g., Vali et al., 2016, 2017; Doekemeijer et al., 2016, 2017; Boersma et al., 2017b). Such a number of states may seem very small for LES simulations, yet is very high for control purposes. Furthermore, $\boldsymbol{q}_k \in \mathbb{R}^O$ includes the system inputs, i.e., the turbine control settings $\gamma_i$ and $C'_{T_i}$ for $i = 1, ..., N_T$. The system outputs $\boldsymbol{z}_k \in \mathbb{R}^M$ are defined by sensors. It can include, among others, flow field measurements ($\boldsymbol{z}_k \subset \boldsymbol{x}_k$) and power measurements. We define the integer $M_{u,v} \in \mathbb{Z}$ with $M_{u,v} \leq M$ as the total number of flow field measurements. The nonlinear functions $f$ and $h$ are the state forward propagation and output equation, respectively.

The computational cost may vary from $0.02$ s for a small wind farm with $N = 3 \cdot 10^3$ states (e.g., a 2 by 1 wind farm in Doekemeijer et al. (2017)), to $1.2$ s for $N = 1 \cdot 10^5$ states for medium-sized wind farms (e.g., a 3 by 3 wind farm in Boersma et al. (2017b)), for a single time-step forward simulation on a single desktop CPU core. This computational complexity is what motivates the use of time-efficient estimation algorithms in the work at hand, and time-efficient predictive control methods for optimization in related work (Vali et al., 2017). In this work, the limits of computational cost are explored to maximize model accuracy while still allowing real-time control.

## 3 Online model calibration

Due to the limited accuracy of surrogate wind farm models, and due to the many uncertainties in the environment, surrogate models often yield predictions with significant uncertainty of the wind flow and power capture inside a wind farm. Since control algorithms largely rely on such predictions, this may suppress gains or even lead to losses inside a wind farm. Unfortu-



nately, higher-fidelity models are computationally prohibitively expensive for control applications. Hence, rather, lower-fidelity surrogate models are calibrated online using readily available measurement equipment.

In this section, first the challenges for real-time model calibration for the surrogate "WFSim" model described in Section 2 will be highlighted in Section 3.1. Secondly, a mathematical framework for recursive model state estimation will be presented in Section 3.2. Thirdly, a number of state estimation algorithms are presented in Sections 3.3 to 3.6, building up from the industry standard to the state of the art in the literature. Finally, a robust, computationally efficient model calibration solution is synthesized in Section 3.7, which allows the simultaneous estimation of the boundary conditions, model parameters, and the model states of WFSim in real time using readily available measurements from the wind farm.

Note that we will henceforth refer to the estimation of $\boldsymbol{x}$ as *state estimation*. The estimation of tuning parameters, such as $\ell_s$ and $c_f$ (Section 2), which are included in the expressions for $f$ and $h$, are considered as *parameter estimation*.

### 3.1 Challenges

Online model calibration for WFSim is challenging for a number of reasons. First of all, the model is nonlinear, and thus the common linear estimation algorithms cannot be used without linearization. While analytical expressions for the linearized surrogate model are available (Boersma et al., 2017b), the absence of linear expressions for the subgrid-scale model and the multiple $max$-, $min$- and $abs$-operators in the nonlinear model limit its accuracy. Secondly, the surrogate model is sensitive to instability when the estimated state sufficiently deviates from the continuity equation in (1). In addition, while state estimation (i.e., estimation of the instantaneous flow field) may prove helpful in short-term forecasting, calibration of additional model parameters is necessary (e.g., the inflow/boundary conditions and the turbulence model) for reliable longer-term forecasting, as will be shown in Section 4. Finally, the surrogate model typically has on the order of $N \sim 10^3 - 10^4$ states, which is extraordinarily high for control applications. Though, real-time estimation is a necessity for real-time model-based closed-loop control, and thus one needs to find a trade-off between accuracy on the one hand, while guaranteeing updates at a low computational cost on the seconds-scale on the other hand.

### 3.2 General formulation

This section details the basics of Kalman filtering, which is the literature standard for state estimation in control. The goal of a Kalman filter (KF) is to estimate the unmeasured states of a dynamical system through noisy measurements. Assumed here is a system (the wind farm) represented mathematically by a discrete-time stochastic state-space model with additive noise,

$$\boldsymbol{x}_{k+1} = f(\boldsymbol{x}_k, \boldsymbol{q}_k) + \boldsymbol{w}_k, \qquad (6)$$

$$\boldsymbol{z}_k = h(\boldsymbol{x}_k, \boldsymbol{q}_k) + \boldsymbol{v}_k, \qquad (7)$$

where $k$ is the time index, $\boldsymbol{x} \in \mathbb{R}^N$ is the unobserved system state (in this case: the flow and pressure fields inside the wind farm), $\boldsymbol{z} \in \mathbb{R}^M$ are the measured outputs of the system (e.g., flow measurements, SCADA data), $f$ describes the forward-in-time state propagation mapping, $h$ describes the output equation from state to measurement, $\boldsymbol{q} \in \mathbb{R}^O$ and $\boldsymbol{w} \in \mathbb{R}^N$ are the controllable inputs and process noise respectively that drive the system dynamics, and $\boldsymbol{v} \in \mathbb{R}^M$ is measurement noise.




Furthermore, we assume $\boldsymbol{w}$ and $\boldsymbol{v}$ to be zero-mean white Gaussian noise with covariance matrices

$$
\mathrm{E}\left[\begin{bmatrix} \boldsymbol{v}_k \\ \boldsymbol{w}_k \end{bmatrix} \begin{bmatrix} \boldsymbol{v}_\ell^T & \boldsymbol{w}_\ell^T \end{bmatrix}\right] = \begin{bmatrix} \boldsymbol{R}_k & \boldsymbol{S}_k^T \\ \boldsymbol{S}_k & \boldsymbol{Q}_k \end{bmatrix} \Delta_{k-\ell}, \quad \text{where} \quad \Delta_{k-\ell} = \begin{cases} 1, & \text{if } k=\ell, \\ 0, & \text{otherwise,} \end{cases} \tag{8}
$$

with E the expectation operator. Estimates of the state $\boldsymbol{x}_k$, denoted by $\hat{\boldsymbol{x}}_{k|k}$, are computed based on measurements from the real system. Here, $\hat{\boldsymbol{x}}_{k|\ell}$ means an estimate of the model's state vector $\boldsymbol{x}$ at time $k$, using all past measurements and inputs $\boldsymbol{\mathcal{Z}}_\ell$,

as

$$
\hat{\boldsymbol{x}}_{k|\ell} = \mathrm{E}\left[\boldsymbol{x}_k | \boldsymbol{\mathcal{Z}}_\ell\right], \quad \text{with} \quad \boldsymbol{\mathcal{Z}}_\ell = \boldsymbol{z}_0, \boldsymbol{z}_1, \boldsymbol{z}_2 \dots \boldsymbol{z}_\ell, \ \boldsymbol{q}_0, \boldsymbol{q}_1, \boldsymbol{q}_2 \dots \boldsymbol{q}_\ell. \tag{9}
$$

State estimates are based on the internal model dynamics and the measurements, weighted according to their respective probability distributions. We aim to find an optimal state estimate, in which optimality is defined as unbiasedness, $\mathrm{E}\left[\boldsymbol{x}_k - \hat{\boldsymbol{x}}_k\right] = 0$, and when the variance of any linear combination of state estimation errors (e.g., the trace of $\mathrm{E}\left[\left(\boldsymbol{x}_k - \hat{\boldsymbol{x}}_k\right)\left(\boldsymbol{x}_k - \hat{\boldsymbol{x}}_k\right)^T\right]$) is

minimized (Verhaegen and Verdult, 2007).

In reality, the assumed model described by $f$ and $h$ always has mismatches with the true system, and the assumptions in (8) often do not hold. Further, the matrices $\boldsymbol{Q}_k$, $\boldsymbol{R}_k$, and $\boldsymbol{S}_k$ are usually not known and rather considered tuning parameters. In practice, the values of $\boldsymbol{R}$ and $\boldsymbol{Q}$ are used to shift the confidence levels between the internal model and the measured values. For $\boldsymbol{R} \ll \boldsymbol{Q}$, estimations will heavily rely on the measurements, while for $\boldsymbol{Q} \ll \boldsymbol{R}$, estimations will mostly rely on the internal

model. Kalman filtering remains one of the most common methods of recursive state estimation, as it has proven successful in many applications. KF algorithms typically consist of two steps, namely:

1. A state and output forecast, including their uncertainties (covariances):

$$
\hat{\boldsymbol{x}}_{k|k-1} = \mathrm{E}\left[f(\boldsymbol{x}_{k-1}, \boldsymbol{q}_{k-1}) + \boldsymbol{w}_{k-1} | \boldsymbol{\mathcal{Z}}_{k-1}\right], \tag{10}
$$

$$
\hat{\boldsymbol{z}}_{k|k-1} = \mathrm{E}\left[h(\boldsymbol{x}_k, \boldsymbol{q}_k) + \boldsymbol{v}_k | \boldsymbol{\mathcal{Z}}_{k-1}\right], \tag{11}
$$

$$
\boldsymbol{P}_{k|k-1}^x = \mathrm{Cov}\left(\boldsymbol{x}_k, \boldsymbol{x}_k | \boldsymbol{\mathcal{Z}}_{k-1}\right) = \mathrm{E}[(\boldsymbol{x}_k - \hat{\boldsymbol{x}}_{k|k-1})(\boldsymbol{x}_k - \hat{\boldsymbol{x}}_{k|k-1})^T], \tag{12}
$$

$$
\boldsymbol{P}_{k|k-1}^z = \mathrm{Cov}\left(\boldsymbol{z}_k, \boldsymbol{z}_k | \boldsymbol{\mathcal{Z}}_{k-1}\right) = \mathrm{E}[(\boldsymbol{z}_k - \hat{\boldsymbol{z}}_{k|k-1})(\boldsymbol{z}_k - \hat{\boldsymbol{z}}_{k|k-1})^T], \tag{13}
$$

$$
\boldsymbol{P}_{k|k-1}^{xz} = \mathrm{Cov}\left(\boldsymbol{x}_k, \boldsymbol{z}_k | \boldsymbol{\mathcal{Z}}_{k-1}\right) = \mathrm{E}[(\boldsymbol{x}_k - \hat{\boldsymbol{x}}_{k|k-1})(\boldsymbol{z}_k - \hat{\boldsymbol{z}}_{k|k-1})^T]. \tag{14}
$$

In (10) and (11), $\hat{\boldsymbol{x}}_{k|\ell}$ and $\hat{\boldsymbol{z}}_{k|\ell}$ are the forecasted system state vector and measurement vector, respectively.

2. An analysis update of the state vector, where the measurements are fused with the internal model:

$$
\boldsymbol{L}_k = \boldsymbol{P}_{k|k-1}^{xz} \cdot \left(\boldsymbol{P}_{k|k-1}^z\right)^{-1} \tag{15}
$$

$$
\hat{\boldsymbol{x}}_{k|k} = \hat{\boldsymbol{x}}_{k|k-1} + \boldsymbol{L}_k \left(\boldsymbol{z}_k - \hat{\boldsymbol{z}}_{k|k-1}\right), \tag{16}
$$

$$
\boldsymbol{P}_{k|k}^x = \mathrm{Cov}\left(\boldsymbol{x}_k, \boldsymbol{x}_k | \boldsymbol{\mathcal{Z}}_k\right) = \boldsymbol{P}_{k|k-1}^x - \boldsymbol{L}_k \boldsymbol{P}_{k|k-1}^z \boldsymbol{L}_k^T. \tag{17}
$$

Here, $\left(\boldsymbol{P}_{k|k-1}^z\right)^{-1}$ in (15) is the pseudo-inverse of $\boldsymbol{P}_{k|k-1}^z$, since this matrix is not necessarily invertible.



As can be seen in (15), a trade-off is made between the measured quantities and the surrogate model using the covariance terms as weights.

### 3.3 Linear Kalman filter

Traditionally, state estimation for linear dynamic models is done using the linear Kalman filter (KF) (Kalman, 1960). In the idealized situation where: 1) the assumptions on noise in (8) hold, 2) the surrogate model $f$ and $h$ perfectly match reality, and 3) $f$ and $h$ are linear in $\boldsymbol{x}$ and $\boldsymbol{q}$, with

$$f(\boldsymbol{x}_k, \boldsymbol{q}_k) = \boldsymbol{A}_k \boldsymbol{x}_k + \boldsymbol{B}_k \boldsymbol{q}_k,$$
$$h(\boldsymbol{x}_k, \boldsymbol{q}_k) = \boldsymbol{C}_k \boldsymbol{x}_k + \boldsymbol{D}_k \boldsymbol{q}_k,$$

where $\boldsymbol{A}_k$, $\boldsymbol{B}_k$, $\boldsymbol{C}_k$, $\boldsymbol{D}_k$ are the (possibly time-varying) matrices of the state-space system, then the linear KF is optimal in the sense that it provides unbiased estimates, $\mathrm{E}(\boldsymbol{x}_k) = \hat{\boldsymbol{x}}_k$, with minimal mean-square error (the trace of $\boldsymbol{P}^x_{k|k}$ is minimized). For (10) to (17), one can derive that in the linear case,

$$\boldsymbol{P}^x_{k|k-1} = \boldsymbol{A}_{k-1} \boldsymbol{P}^x_{k-1|k-1} \boldsymbol{A}^T_{k-1} + \boldsymbol{Q}_{k-1}, \tag{18}$$

$$\boldsymbol{P}^z_{k|k-1} = \boldsymbol{C}_k \boldsymbol{P}^x_{k|k-1} \boldsymbol{C}^T_k + \boldsymbol{R}_k, \tag{19}$$

$$\boldsymbol{P}^{xz}_{k|k-1} = \boldsymbol{P}^x_{k|k-1} \boldsymbol{C}^T_k + \boldsymbol{S}_k. \tag{20}$$

If any of the three criteria is not met, optimality of the KF is lost. While points 1 and 2 are practically never met, good results are often still achieved. The crux lies with point 3. Namely, the traditional KF cannot deal with nonlinearity in the surrogate model ($\hat{f}$ and/or $\hat{h}$).

### 3.4 Extended Kalman filter (ExKF)

Linearization of the surrogate model is the most straight-forward solution to the issue of nonlinearity in $f(\boldsymbol{x}, \boldsymbol{q})$ and $h(\boldsymbol{x}, \boldsymbol{q})$. This is what the Extended KF (ExKF) does. Here, the surrogate model is linearized around some point $(\boldsymbol{x}^{\mathrm{lin}}, \boldsymbol{q}^{\mathrm{lin}})$ w.r.t. $\boldsymbol{x}$ and $\boldsymbol{q}$ at every timestep $k$:

$$f(\boldsymbol{x}_k, \boldsymbol{q}_k) \approx \underbrace{\left.\frac{\partial f(\boldsymbol{x}, \boldsymbol{q})}{\partial \boldsymbol{x}}\right|_{\boldsymbol{x}^{\mathrm{lin}}, \boldsymbol{q}^{\mathrm{lin}}}}_{\boldsymbol{A}_k} \left(\boldsymbol{x}_k - \boldsymbol{x}^{\mathrm{lin}}\right) + \underbrace{\left.\frac{\partial f(\boldsymbol{x}, \boldsymbol{q})}{\partial \boldsymbol{q}}\right|_{\boldsymbol{x}^{\mathrm{lin}}, \boldsymbol{q}^{\mathrm{lin}}}}_{\boldsymbol{B}_k} \left(\boldsymbol{q}_k - \boldsymbol{q}^{\mathrm{lin}}\right),$$

$$h(\boldsymbol{x}_k, \boldsymbol{q}_k) \approx \underbrace{\left.\frac{\partial h(\boldsymbol{x}, \boldsymbol{q})}{\partial \boldsymbol{x}}\right|_{\boldsymbol{x}^{\mathrm{lin}}, \boldsymbol{q}^{\mathrm{lin}}}}_{\boldsymbol{C}_k} \left(\boldsymbol{x}_k - \boldsymbol{x}^{\mathrm{lin}}\right) + \underbrace{\left.\frac{\partial h(\boldsymbol{x}, \boldsymbol{q})}{\partial \boldsymbol{q}}\right|_{\boldsymbol{x}^{\mathrm{lin}}, \boldsymbol{q}^{\mathrm{lin}}}}_{\boldsymbol{D}_k} \left(\boldsymbol{q}_k - \boldsymbol{q}^{\mathrm{lin}}\right).$$

Using the linearized system matrices $\boldsymbol{A}_k$, $\boldsymbol{B}_k$, $\boldsymbol{C}_k$, $\boldsymbol{D}_k$, one can directly apply (10) to (17) for state estimation, where (18) to (20) become approximations instead of equalities. Fundamentally, in the ExKF, the state is assumed to have a Gaussian probability distribution. This variable is propagated through the linearized system dynamics, yielding a posterior distribution which is also



Gaussian. Hence, the ExKF can be considered a first-order approximation of the true state probability distribution. Optimality is not guaranteed, and this lower-order approximation can even lead to divergence for some models. Though, the ExKF has shown success in academia and industry (Wan and Van Der Merwe, 2000).

As described in Section 3.1, model linearization is troublesome. Furthermore, for surrogate models with many states such as WFSim, the ExKF has an additional challenge: computational complexity. The operation in (15) includes a matrix inversion with a computational complexity of $\mathcal{O}(M^3)$, and (18) includes two matrix multiplications each with a computational complexity of $\mathcal{O}(N^3)$. As there are significantly fewer measurements than states ($M \ll N$) for the problem at hand, (18) dominates the computational cost. For example, a mesh in WFSim with $50 \times 25$ cells yields a state vector size of $N = 3 \cdot 10^3$, and (18) contributes to about $80 - 90\%$ of the computational cost for the entire KF cycle, with a CPU time on the order of $1 \cdot 10^1$ s, about one order of magnitude too large for online model calibration. To reduce computational cost in the ExKF, the surrogate model and/or the covariance matrix $P$ have to be simplified. This is not further explored here. Instead, two KF approaches will be explored that use the nonlinear system directly for forecasting and analysis updates. Doing so, we circumvent the problems with linearization, and additionally better maintain the true covariance of the system state.

### 3.5 The Unscented Kalman filter (UKF)

The Unscented Kalman filter (UKF) relies on the so-called "unscented transformation" (UT) to estimate the means and covariance matrices described by (10) to (14). The conditional state probability distribution of $\boldsymbol{x}_k$ knowing $\boldsymbol{\mathcal{Z}}_k$ is again assumed to be Gaussian. In the UKF, firstly a number of sigma points (also referred to as "particles") are generated such that their mean is equal to $\hat{\boldsymbol{x}}_{k|k}$ and their covariance is equal to $\mathrm{Cov}\left(\boldsymbol{x}_k, \boldsymbol{x}_k\right)$. Secondly, each particle is propagated through the nonlinear system dynamics ($f$, $h$). Thirdly, the mean and covariance of the forecasted state probability distribution is again approximated by a weighted mean of these forecasted sigma points (Wan and Van Der Merwe, 2000).

Mathematically, we define the $i^{\text{th}}$ particle as $\boldsymbol{\psi}_{k|\ell}^i \in \mathbb{R}^N$, which is a realization of the condition probability distribution of $\boldsymbol{x}_k$ given $\boldsymbol{\mathcal{Z}}_\ell$. The UKF follows a very similar forecast and analysis update approach as the traditional KF in (10) to (17), yet applied to a finite set of particles (Wan and Van Der Merwe, 2000).

1. For the forecast step, a particle-based approach is taken.

    (i) A total of $Y = 2N + 1$ particles are (re)sampled to capture the mean and covariance of the conditional state probability distribution $p[\boldsymbol{x}_{k-1}|\boldsymbol{\mathcal{Z}}_{k-1}]$, by

$$\boldsymbol{\psi}_{k-1|k-1}^i = \begin{cases} \overline{\boldsymbol{\psi}}_{k-1|k-1} & \text{for } i = 1, \\ \overline{\boldsymbol{\psi}}_{k-1|k-1} + \left(\sqrt{(N+\lambda)\cdot\boldsymbol{P}_{k-1|k-1}^{\boldsymbol{x}}}\right)_i & \text{for } i = 2, ..., N+1, \\ \overline{\boldsymbol{\psi}}_{k-1|k-1} - \left(\sqrt{(N+\lambda)\cdot\boldsymbol{P}_{k-1|k-1}^{\boldsymbol{x}}}\right)_{i-N-1} & \text{for } i = N+2, ..., Y, \end{cases} \tag{21}$$

    where $\lambda = \alpha^2(N+\kappa) - N$ is a scaling parameter, where $\alpha$ determines the spread of the particles around the mean, and $\kappa$ is a secondary scaling parameter typically set to 0 (Wan and Van Der Merwe, 2000). The weight of each particle's mean $\boldsymbol{w}_{\text{mean}}^i$ and covariance $\boldsymbol{w}_{\text{cov.}}^i$ is given by





$$
\boldsymbol{w}^i_{\text{mean}} = \begin{cases} \lambda(N+\lambda)^{-1} & \text{for } i = 1, \\ \frac{1}{2}(N+\lambda)^{-1} & \text{otherwise,} \end{cases} \qquad \boldsymbol{w}^i_{\text{cov.}} = \begin{cases} \lambda(N+\lambda)^{-1} + (1 - \alpha^2 + \beta) & \text{for } i = 1, \\ \frac{1}{2}(N+\lambda)^{-1} & \text{otherwise,} \end{cases}
$$

where $\beta$ is used to incorporate prior knowledge on the probability distribution. In this work, $\beta = 2$ is assumed, which is stated to be optimal for Gaussian distributions (Wan and Van Der Merwe, 2000).

(ii) Each particle is propagated forward in time using the expectation of the nonlinear model, as

$$
\begin{aligned}
\boldsymbol{\psi}^i_{k|k-1} &= f(\boldsymbol{\psi}^i_{k-1|k-1}, \boldsymbol{q}_{k-1}) && \text{for } i = 1,...,Y, \\
\boldsymbol{\zeta}^i_{k|k-1} &= h(\boldsymbol{\psi}^i_{k|k-1}, \boldsymbol{q}_k) && \text{for } i = 1,...,Y,
\end{aligned} \tag{22}
$$

where $\boldsymbol{\zeta}^i_{k|\ell}$ is defined as the system output corresponding to the particle $\boldsymbol{\psi}^i_{k|\ell}$.

(iii) The expected state $\overline{\boldsymbol{\psi}}$ and expected output $\overline{\boldsymbol{\zeta}}$ are calculated as

$$
\begin{aligned}
\hat{\boldsymbol{x}}_{k|k-1} \approx \overline{\boldsymbol{\psi}}_{k|k-1} &= \sum_{i=1}^{Y} \left( \boldsymbol{w}^i_{\text{mean}} \cdot \boldsymbol{\psi}^i_{k|k-1} \right), \\
\hat{\boldsymbol{z}}_{k|k-1} \approx \overline{\boldsymbol{\zeta}}_{k|k-1} &= \sum_{i=1}^{Y} \left( \boldsymbol{w}^i_{\text{mean}} \cdot \boldsymbol{\zeta}^i_{k|k-1} \right),
\end{aligned} \tag{23}
$$

and the covariance matrices are (re-)estimated from the forecasted ensemble by

$$
\boldsymbol{P}^x_{k|k-1} = \sum_{i=1}^{Y} \left( \boldsymbol{w}^i_{\text{cov.}} \left( \boldsymbol{\psi}^i_{k|k-1} - \overline{\boldsymbol{\psi}}_{k|k-1} \right) \left( \boldsymbol{\psi}^i_{k|k-1} - \overline{\boldsymbol{\psi}}_{k|k-1} \right)^T \right) + \boldsymbol{Q}_{k-1}, \tag{24}
$$

$$
\boldsymbol{P}^z_{k|k-1} = \sum_{i=1}^{Y} \left( \boldsymbol{w}^i_{\text{cov.}} \left( \boldsymbol{\zeta}^i_{k|k-1} - \overline{\boldsymbol{\zeta}}_{k|k-1} \right) \left( \boldsymbol{\zeta}^i_{k|k-1} - \overline{\boldsymbol{\zeta}}_{k|k-1} \right)^T \right) + \boldsymbol{R}_k, \tag{25}
$$

$$
\boldsymbol{P}^{xz}_{k|k-1} = \sum_{i=1}^{Y} \left( \boldsymbol{w}^i_{\text{cov.}} \left( \boldsymbol{\psi}^i_{k|k-1} - \overline{\boldsymbol{\psi}}_{k|k-1} \right) \left( \boldsymbol{\zeta}^i_{k|k-1} - \overline{\boldsymbol{\zeta}}_{k|k-1} \right)^T \right) + \boldsymbol{S}_k. \tag{26}
$$

2. For the analysis step, one can apply the same equations as in (15) to (17).

The UKF has been shown to consistently outperform the ExKF in terms of accuracy, since it uses the nonlinear model for model forecasting and covariance propagation. However, this does come at a cost. Namely, $Y = 2N + 1$ particles are required to capture the mean and covariance of the conditional state probability distribution. This implies that $2N + 1$ function evaluations are required for each UKF update. Even for a small wind farm in WFSim, $N = 3 \cdot 10^3$, one function evaluation takes approximately 0.02 s. This means that a lower limit on the computational cost of the UKF algorithm is $1 \cdot 10^2$ s on a single core for a single-timestep forward simulation ($k \to k+1$). While (22) can easily be parallelized, computational complexity remains troublesome, especially for larger wind farms. Rather, a more computationally efficient particle-based KF algorithm is investigated. This is the Ensemble Kalman filter described in Section 3.6.





### 3.6 The Ensemble Kalman filter (EnKF)

The Ensemble Kalman filter (EnKF) (Evensen, 2003) is very similar to the UKF in that it relies on a finite number of realizations (the "sigma points" or "particles" in the UKF) to approximate the mean and covariance of the conditional probability distribution of $\boldsymbol{x}_k$ knowing $\boldsymbol{\mathcal{Z}}_k$. However, whereas the UKF relies on a systematic way of distributing the particles such that

the mean and covariance of the conditional probability distribution $p\left[\boldsymbol{x}_k | \boldsymbol{\mathcal{Z}}_k\right]$ are equal to that of the particles, the EnKF relies on random realizations, without guarantees that the mean and covariance are captured accurately. Though, the EnKF has been shown to work well in a number of applications, with typically far fewer particles than states, i.e., $Y \ll N$ (e.g., Houtekamer and Mitchell, 2005; Gillijns et al., 2006). The forecast and update step are very similar to that of the UKF, namely:

1. In the UKF the particles are redistributed at every timestep, in contrast to the EnKF. Rather, the EnKF propagates the
particles forward without redistribution. We define the $i^{\text{th}}$ particle as $\boldsymbol{\psi}_{k|\ell}^i \in \mathbb{R}^N$, which is a realization of the conditional probability distribution $p\left[\boldsymbol{x}_k | \boldsymbol{\mathcal{Z}}_\ell\right]$. The forecast step is:

   (i) Each particle is propagated forward in time using the nonlinear system dynamics, and with the realizations of noise terms $\boldsymbol{w}$ and $\boldsymbol{v}$ denoted by $\hat{\boldsymbol{w}}_{k-1}^i \in \mathbb{R}^N$ and $\hat{\boldsymbol{v}}_k^i \in \mathbb{R}^M$, respectively.

$$
\begin{aligned}
\boldsymbol{\psi}_{k|k-1}^i &= f(\boldsymbol{\psi}_{k-1|k-1}^i, \boldsymbol{q}_{k-1}) + \hat{\boldsymbol{w}}_{k-1}^i && \text{for} \quad i = 1, ..., Y, \\
\boldsymbol{\zeta}_{k|k-1}^i &= h(\boldsymbol{\psi}_{k|k-1}^i, \boldsymbol{q}_k) + \hat{\boldsymbol{v}}_k^i && \text{for} \quad i = 1, ..., Y.
\end{aligned}
\tag{27}
$$

(ii) The expected state and output are calculated identically as in the UKF using (23) with $\boldsymbol{w}_{\text{mean}}^i = (Y-1)^{-1}$. The covariance matrices are (re-)estimated from the forecasted ensemble, by

$$
\boldsymbol{P}_{k|k-1}^z = \frac{1}{Y-1} \sum_{i=1}^Y \left( \left( \boldsymbol{\zeta}_{k|k-1}^i - \overline{\boldsymbol{\zeta}}_{k|k-1} \right) \left( \boldsymbol{\zeta}_{k|k-1}^i - \overline{\boldsymbol{\zeta}}_{k|k-1} \right)^T \right),
\tag{28}
$$

$$
\boldsymbol{P}_{k|k-1}^{xz} = \frac{1}{Y-1} \sum_{i=1}^Y \left( \left( \boldsymbol{\zeta}_{k|k-1}^i - \overline{\boldsymbol{\zeta}}_{k|k-1} \right) \left( \boldsymbol{\psi}_{k|k-1}^i - \overline{\boldsymbol{\psi}}_{k|k-1} \right)^T \right).
\tag{29}
$$

2. For the analysis step, one applies (15) to determine the Kalman gain $\boldsymbol{L}_k$. Then, each particle is updated individually, as

$$
\boldsymbol{\psi}_{k|k}^i = \boldsymbol{\psi}_{k|k-1}^i + \boldsymbol{L}_k \left( \boldsymbol{z}_k - \boldsymbol{\zeta}_{k|k-1}^i \right) \qquad \text{for} \quad i = 1, ..., Y.
\tag{30}
$$

Note that, in contrast to the ExKF and the UKF, the state covariance matrix $\boldsymbol{P}^x$ (see (12) and (17)) need not be calculated explicitly in the EnKF. This, in combination with the small number of particles $Y \ll N$, is what makes the EnKF computationally superior to the UKF (and often also computationally superior to the ExKF). However, this reduction in computational complexity comes at a price. The disadvantages of the EnKF are discussed in the next section.

### 3.6.1 Challenges in the EnKF for small number of particles

The caveat to representing the conditional state probability distribution with fewer particles than states, $Y \ll N$, is the formation of inbreeding and long-range spurious correlations (Petrie, 2008). The former, inbreeding, is defined as a situation where

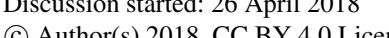



the state error covariance matrix $\boldsymbol{P}^x$ is consistently underestimated, leading to state estimates that incorrectly rely more on the internal model. One straight-forward method to address this is called "covariance inflation", in which $\boldsymbol{P}^x$ (or rather, the ensemble from which $\boldsymbol{P}^x$ is calculated) is scaled (the ensemble is "inflated") to correct for the underestimated state uncertainty (Petrie, 2008). Mathematically, this is achieved by applying

$$\boldsymbol{\psi}_{k|k-1}^i = \overline{\boldsymbol{\psi}}_{k|k-1} + r\left(\boldsymbol{\psi}_{k|k-1}^i - \overline{\boldsymbol{\psi}}_{k|k-1}\right) \qquad \text{for} \quad i = 1,...,Y, \tag{31}$$

before the analysis step, with $r \in \mathbb{R}$ the inflation factor, typically with a value of $1.01 - 1.25$.

The latter problem, long-range spurious correlations, can be better visualized in Fig. 3. In particle-based approaches, the

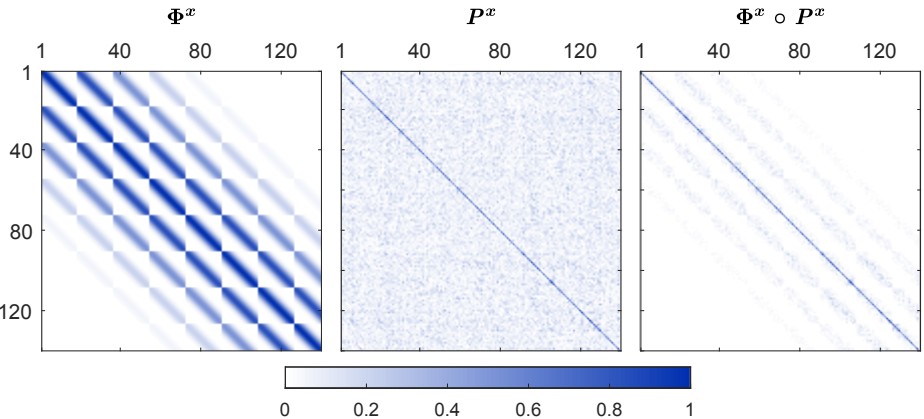

**Figure 3.** Long-range spurious correlations arise in the case where a covariance matrix is described by a small number of particles. Using physical knowledge of the system, these undesired correlations can be corrected. $\boldsymbol{\Phi}^x$ is the localization matrix. Applying localization, the covariance of physically nearby states are multiplied with a value close to 1, and the covariance of physically distant states are multiplied with a value close to 0. In our example case, this results in the localized covariance matrix $\boldsymbol{\Phi}^x \circ \boldsymbol{P}^x$, where $\circ$ is the element-wise product.

covariance terms cannot be captured exactly. This may lead to the formation of small yet nonzero covariance terms between states and outputs which, in reality, are uncorrelated. Then, these states will be adapted according to uncorrelated measure-

ments, which can lead to the drift of unobservable states (states for which no information is available). This state drift can build up and lead to instability of the surrogate model. Increasing the number of particles is the most straight-forward solution to this problem, but comes at a huge computational cost. A better alternative is "covariance localization", where physical knowledge of the states and measurements is used to steer the sample-based covariance matrices. Recall that in the surrogate model of Section 2, the model states are the velocity and pressure terms inside the wind farm at a physical location. Define that the $i^{\text{th}}$

state entry $(\boldsymbol{x}_k)_i$ belongs to a physical location in the farm $\boldsymbol{s}_i$. Then, looking at an arbitrary state covariance term $(i,j)$,

$$\left(\boldsymbol{P}_{k|k-1}^x\right)_{i,j} = E\left[\left((\boldsymbol{x}_k)_i - (\hat{\boldsymbol{x}}_{k|k-1})_i\right)\left((\boldsymbol{x}_k)_j - (\hat{\boldsymbol{x}}_{k|k-1})_j\right)^T\right],$$

we define the physical distance between these two states as $\Delta s_{i,j} = ||\boldsymbol{s}_i - \boldsymbol{s}_j||_2$. Now, we introduce a weighting factor into our covariance matrices by multiplying physically distant states with a value close to 0, and multiplying physically nearby





states with a value close to 1. A popular choice for such a weighting function is Gaspari-Cohn's fifth-order discretization of a Gaussian distribution (Gaspari and Cohn, 1999), given by

$$
\phi(c_{i,j}) = \begin{cases} -\frac{1}{4}c_{i,j}^5 + \frac{1}{2}c_{i,j}^4 + \frac{5}{8}c_{i,j}^3 - \frac{5}{3}c_{i,j}^2 + 1 & \text{if } 0 \leq c_{i,j} \leq 1, \\ \frac{1}{12}c_{i,j}^5 - \frac{1}{2}c_{i,j}^4 + \frac{5}{8}c_{i,j}^3 + \frac{5}{3}c_{i,j}^2 - 5c_{i,j} + 4 - \frac{2}{3}\frac{1}{c_{i,j}} & \text{if } 1 < c_{i,j} \leq 2, \\ 0 & \text{otherwise,} \end{cases}
\tag{32}
$$

with $c_{i,j} = \frac{||\Delta s_{i,j}||_2}{L}$ a normalized distance measure, with $L$ the cut-off distance. Applying (32) for the covariance matrices
$\boldsymbol{P}_{k|k-1}^z$ and $\boldsymbol{P}_{k|k-1}^{xz}$ (note that the state covariance matrix $\boldsymbol{P}_{k|k-1}^x$ is not calculated explicitly in the EnKF, but could be calculated similarly), we can define the localization matrices

$$
\boldsymbol{\Phi}^z = \begin{bmatrix} \phi(c_{1,1}^z) & \cdots & \cdots\phi(c_{1,N}^z) \\ \vdots & \ddots & \\ \phi(c_{N,1}^z) & & \phi(c_{N,N}^z) \end{bmatrix}, \qquad \boldsymbol{\Phi}^{xz} = \begin{bmatrix} \phi(c_{1,1}^{xz}) & \cdots & \cdots\phi(c_{1,N}^{xz}) \\ \vdots & \ddots & \\ \phi(c_{N,1}^{xz}) & & \phi(c_{N,N}^{xz}) \end{bmatrix},
$$

where $c_{i,j}^z$ is the normalized distance between two measurements $i$ and $j$, and $c_{i,j}^{xz}$ is the normalized distance between state $i$ and measurement $j$, respectively. Finally, localization and inflation can be incorporated into (28) and (29) by

$$
\boldsymbol{P}_{k|k-1}^z = \boldsymbol{\Phi}^z \circ \frac{1}{Y-1}\sum_{i=1}^{Y}\left( \left(\boldsymbol{\zeta}_{k|k-1}^i - \overline{\boldsymbol{\zeta}}_{k|k-1}\right)\left(\boldsymbol{\zeta}_{k|k-1}^i - \overline{\boldsymbol{\zeta}}_{k|k-1}\right)^T \right),
\tag{33}
$$

$$
\boldsymbol{P}_{k|k-1}^{xz} = \sqrt{r}\cdot\boldsymbol{\Phi}^{xz} \circ \frac{1}{Y-1}\sum_{i=1}^{Y}\left( \left(\boldsymbol{\zeta}_{k|k-1}^i - \overline{\boldsymbol{\zeta}}_{k|k-1}\right)\left(\boldsymbol{\psi}_{k|k-1}^i - \overline{\boldsymbol{\psi}}_{k|k-1}\right)^T \right),
\tag{34}
$$

where $\circ$ is the element-wise product (Hadamard) of the two matrices. The improvement in terms of computational efficiency and estimation performance is displayed in Fig. 4. A very significant increase in performance is shown, especially for smaller numbers of particles. This is in agreement with what was seen in previous work (Doekemeijer et al., 2017). Furthermore,
performance is much more consistent. Additionally, note that there is no increase in computational cost, as the covariance matrices are sparsified, leading to a cost reduction in the calculation of (15), which makes up for the extra operations of (33) and (34). Also, note that the localization matrices are time-invariant and can be calculated offline.

### 3.7   Synthesizing an online model calibration solution

As mentioned in Section 3.1, parameter estimation may be even of a higher importance than state estimation for longer-term
forecasting. Parameter estimation is achieved by extending the state vector with (a subset of) the model parameters. In this work, the model parameter $\ell_s$ (turbulence mixing length factor) is concatenated to the state vector. Higher values of $\ell_s$ lead to more mixing behind the turbines, yielding more wake recovery, making the calibration solution adaptable to varying turbulence levels. This adds one scalar entry to $\boldsymbol{x}$, which is a negligible addition in terms of computational cost.

The freestream wind speed $U_\infty$ and direction $\phi$ in the wind farm are also estimated in this framework. This is done using
the turbine's wind vane and power generation measurements, following the ideas of Gebraad et al. (2016) and Shapiro et al.




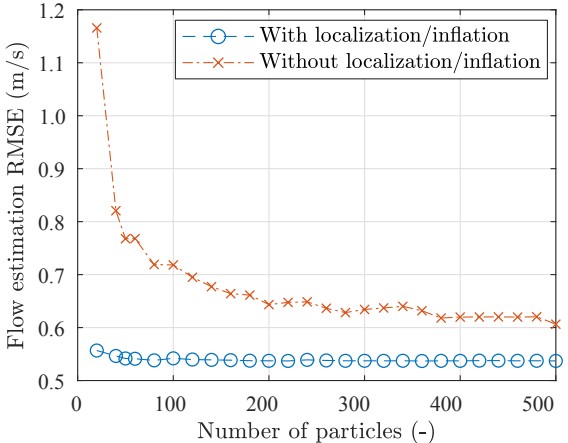
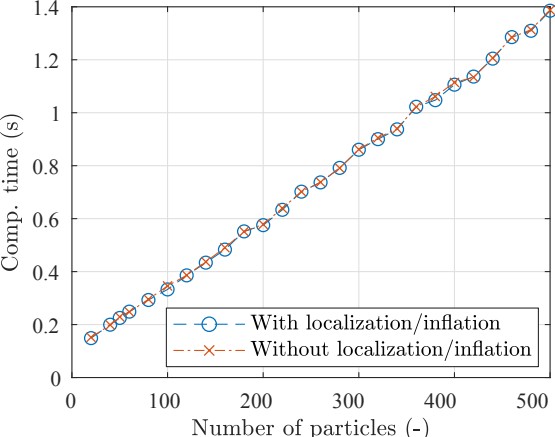

**Figure 4.** This figure shows the estimation performance and computational cost (parallelized, 8 cores) of the EnKF for a range of ensemble sizes, with and without inflation and localization. Great improvement is seen for estimation accuracy, at no additional computational cost. The simulation scenario is described in detail in Section 4.2, and the results presented here are rather meant as an indication.

(2017b). Using the wind vanes, $\phi$ can be calculated as the average of the wind vane measurements. Knowing this, and employing a simple steady-state wake model from the literature (Mittelmeier et al., 2017), the turbines operating in freestream flow can be distinguished from the ones operating in waked flow. Next, define $\amalg \in \mathbb{Z}^{\mathbb{K}}$ as a vector specifying the upstream turbines, with $\mathbb{K} \in \mathbb{Z}$ the total number of turbines operating in freestream. Then, the instantaneous rotor-averaged flow speed

at each turbine's hub can be estimated using the inverse relationship of (5). One wind-farm-wide freestream wind speed $U_\infty$ is then calculated using actuator disk theory. Smoothing results with a low-pass filter action on the average of $U_{\infty_i}$ for each upstream turbine $i$, we obtain

$$c_{u_\infty} \frac{\partial U_\infty}{\partial t} = \frac{1}{\mathbb{K}} \sum_{i \in \amalg} \left( \sqrt[3]{\frac{P_{\text{turb},i}^{\text{meas.}}}{\frac{c_p}{2} \rho A C'_{T_i} \cos(\gamma_i)^3}} \cdot \left(1 + \frac{1}{4} C'_{T_i}\right) \right) - U_\infty, \tag{35}$$

where we used actuator disk theory for the identity

$U_{\infty_i} \approx U_{r_i} \left(1 + 0.25 \cdot C'_{T_i}\right)$, when $\gamma_i \approx 0$.

Furthermore, $c_{u_\infty}$ is the time constant of the first-order low-pass filter, and $P_{\text{turb},i}^{\text{meas.}}$ is the measured instantaneous power capture of turbine $i$. While the assumption $\gamma = 0$ is made here for the calculation of $U_\infty$, research is currently ongoing on how to best incorporate the effects of turbine yaw ($\gamma \neq 0$) into the definition of $C'_T$.

An important remark is that this methodology for the estimation of $U_\infty$ relies solely on power measurements, and therefore

only works for below-rated conditions. For estimation of $U_\infty$ in above-rated conditions, one may, for example, require the implementation of a wind speed estimator on each individual turbine, from which the local wind speed in front of each turbine can be estimated, as demonstrated by Simley and Pao (2016).



Combining these elements yields an efficient, modular, and accurate model calibration solution for low- and medium-fidelity dynamic wind farm models. The natural model states are estimated using SCADA and/or LIDAR data inside a wind farm, of which the former is readily available, and the latter becoming more popular. State estimation greatly improves short-term forecasting, important for the small timescales involved in active power control for wind farms. Furthermore, model parameters

can be estimated online in parallel with the states, and is required for accurate long-term forecasting and important for active power control at lower frequencies. Additionally, the freestream conditions (boundary conditions in our surrogate model, see Section 2.4) are estimated using readily available SCADA data.

This control solution is implemented in MATLAB, but leverages the numerically efficient precompiled solvers for model propagation. Furthermore, the forecasting step of (27) is parallelized, making the EnKF easily scalable up to $Y$ cores. This, in

combination with covariance localization and inflation, makes the EnKF orders of magnitude faster than existing estimation algorithms, while competing with the UKF in terms of accuracy.

## 4  Results

In this section, the model calibration solution detailed in Section 3 will be validated using high-fidelity simulation data. First, the simulation tool used to generate the high-fidelity validation data will be described in Section 4.1. Then, a two-turbine and

a nine-turbine simulation case are presented in Sections 4.2 and 4.3, respectively.

Note that for the all presented results, pressure terms are ignored in the state vector, as they appeared unnecessary for the estimation of flow fields and powers in previous work (Doekemeijer et al., 2017). Furthermore, for simplicity and due to lack of information about the true system, the process and measurement noise will be assumed to be uncorrelated, i.e., $\boldsymbol{S}_k = 0$, and $\boldsymbol{Q}_k$ and $\boldsymbol{R}_k$ are assumed to be time-invariant and diagonal.

### 4.1  SOWFA

High-fidelity simulation data is generated using the Simulator fOr Wind Farm Applications (SOWFA), developed by the National Renewable Energy Laboratory (NREL). This wind farm model provides highly accurate flow data at a fraction of the cost of field tests. SOWFA solves the filtered, three-dimensional, unsteady, incompressible Navier-Stokes equations over a finite temporal and spatial mesh. SOWFA is a large-eddy simulation solver, meaning that larger scale dynamics are resolved directly,

but turbulent structures smaller than the discretization are approximated using subgrid-scale models to suppress computational cost. Coriolis and geostrophic forcing terms are included in SOWFA (Churchfield et al., 2016). The turbine rotor is modeled using an actuator line representation as derived from Sorensen and Shen (2002). In the actuator line model (ALM), the rotor blades are discretized spatially along their radial lines, where lift and drag forces are determined based on the incoming flow angle, flow velocity, and blade (airfoil) geometry (Fleming et al., 2015).

SOWFA has previously been used for lower-fidelity model validation, controller testing, and to study the aerodynamics in wind farms (e.g., Fleming et al., 2015, 2016, 2017a; Gebraad et al., 2016, 2017). The interested reader is referred to Churchfield et al. (2012) for a more in-depth description of SOWFA and LES solvers in general.





### 4.2 2-turbine ALM with turbulent inflow

In this section, a two-turbine wind farm is simulated to highlight the need for state and model parameter estimation, and to motivate the use for the EnKF. This simple wind farm contains two NREL 5-MW baseline turbines with $D = 126.4$ m, separated five turbine diameters apart. This LES simulation has been used before in the literature and was described in more detail in Annoni et al. (2016b). Several important simulation properties are listed in Table 1 for SOWFA and WFSim. The effect of the turbulence intensity on the wake dynamics in SOWFA is captured in WFSim through its mixing-length turbulence model. In these simulations, WFSim is purposely initialized with a too low value for $\ell_s$ in order to represent the realistic situation of a model mismatch. The remaining tuning parameters in WFSim were chosen such that a weighted-sum cost function of the power and flow errors was minimized.

**Table 1.** Overview of several settings for the SOWFA and the WFSim 2-turbine wind farm simulation.

| Variable | Symbol | SOWFA | WFSim |
|---|---|---|---|
| Domain size | - | $3.0\text{km} \times 3.0\text{km} \times 1.0\text{km}$ | $1.9\text{km} \times 0.80\text{km}$ |
| Number of states | $N$ | $\mathcal{O}(10^8) - \mathcal{O}(10^9)$ | $3.2 \cdot 10^3$ |
| Cell size near rotors | - | $3\text{m} \times 3\text{m} \times 3\text{m}$ | $38\text{m} \times 33\text{m}$ |
| Cell size outer regions | - | $12\text{m} \times 12\text{m} \times 12\text{m}$ | $38\text{m} \times 33\text{m}$ |
| Rotor model | - | ALM | ADM ($c_f = 1.4$, $c_p = 0.95$) |
| Inflow wind speed | $U_\infty$ | 8.0 m/s | 8.0 m/s |
| Atmospheric turbulence | - | Turbulent inflow, $\text{TI}_\infty = 5.0\%$ | $d' = 1.8 \cdot 10^2$ m, $d = 6.1 \cdot 10^2$ m, $\ell_s = 1.8 \cdot 10^{-2}$ |

Firstly, the three KF variants will be compared for state estimation in Section 4.2.1. Secondly, in Section 4.2.2, estimation using different information (sensor) sources is compared. Thirdly, the strength of simultaneous state-parameter estimation is displayed in Section 4.2.3.

### 4.2.1 A comparison of the KF variants for state estimation

First, the performance of the ExKF, UKF, EnKF, and the case without estimation (denoted as open-loop, or "OL") are compared for the two-turbine simulation case of Table 1. This simulation only focuses on estimation of the model states, not the model parameter $\ell_s$. Flow measurements downstream of each turbine are assumed (e.g., using LiDAR), their locations denoted as red dots in Fig. 5, which is about $2\%$ of the full to-be-estimated state space. These measurements are artificially disturbed by zero-mean white noise with $\sigma = 0.10$ m/s. The KF settings are listed in Tables 2 and 3. The KF covariance matrices were obtained through an iterative tuning process in previous work (Doekemeijer et al., 2017) with minor adjustments, to simulate performance for untrained data. Figure 5 shows state (flow field) estimation of the three KF variants for two time instants, $t = 300$ s and $t = 700$ s. In this figure, $(\Delta \boldsymbol{u})_\bullet \in \mathbb{R}^{N_u}$ is defined as the error between the estimated and true longitudinal flow



**Table 2.** Covariance settings for the KF variants, with $I_\bullet$ the $\mathbb{R}^{\bullet \times \bullet}$ identity matrix. The full cov. matrices are diagonal concatenations of the entries. For example, $P_0$ is $\mathbf{diag}\,(P_{0,u},\ P_{0,v})$ and $\mathbf{diag}\,(P_{0,u},\ P_{0,v},\ P_{0,\ell_s})$ for state-only and dual estimation, respectively.

| Variable | Symbol | Units | Value |
|---|---|---|---|
| Init. state error cov. of $u_k$ | $P_{0,u}$ | (m/s)$^2$ | $1.0 \cdot 10^{-1} \cdot I_{N_u}$ |
| Init. state error cov. of $v_k$ | $P_{0,v}$ | (m/s)$^2$ | $1.0 \cdot 10^{-1} \cdot I_{N_v}$ |
| Init. state error cov. of $\ell_{s_k}$ | $P_{0,\ell_s}$ | – | $5.0 \cdot 10^{-1}$ |
| Model error cov. of $u_k$ | $Q_{0,u}$ | (m/s)$^2$ | $1.0 \cdot 10^{-2} \cdot I_{N_u}$ |
| Model error cov. of $v_k$ | $Q_{0,v}$ | (m/s)$^2$ | $1.0 \cdot 10^{-4} \cdot I_{N_v}$ |
| Model error cov. of $\ell_{s_k}$ | $Q_{0,\ell_s}$ | – | $1.0 \cdot 10^{-4}$ |
| Meas. error cov. of flow | $R_{u,v}$ | (m/s)$^2$ | $1.0 \cdot 10^{-2} \cdot I_{M_{u,v}}$ |
| Meas. error cov. of $P$ | $R_P$ | (W)$^2$ | $1.0 \cdot 10^{8} \cdot I_{N_T}$ |

**Table 3.** Choice of tuning parameters for the KF variants, for both the 2-turbine and 9-turbine simulation case. Note that the ExKF does not support power measurements nor parameter estimation due to the lack of linearization, and does not have any additional tuning parameters. In terms of computational cost: simulations were run on a single node using 8 cores in parallel.

| Variable | 2-turb. ExKF | 2-turb. UKF | 2-turb. EnKF | 9-turb. EnKF |
|---|---|---|---|---|
| Number of particles, $Y$ | – | 4275 | 50 | 50 |
| Tuning parameters | – | $\alpha$   1.0 <br> $\beta$   2.0 <br> $\kappa$   0 | $L$   131 m <br> $r$   1.025 | $L$   131 m <br> $r$   1.025 |
| Comp. cost/it. | 16.2 s | 14.0 s | 0.25 s | 1.2 s |

velocities in the field, given by

$$(\Delta u)_\bullet = |u_\bullet - u_{\mathrm{SOWFA}}|.$$

Looking at Fig. 5, the open-loop estimations are accurate for the unwaked and single waked flow, yet are lacking in the situation of two overlapping wakes, for which the KFs correct. There is no significant difference in accuracy between the
5 different KF variants, yet they differ by two orders of magnitude in computational cost (Table 3).

### 4.2.2 A comparison of sensor configurations

Previous results (Doekemeijer et al., 2016, 2017) have relied on flow measurements for state estimation. However, in existing wind farms, such measurements are typically not available. Rather, readily available SCADA data should be used for the purpose of model calibration. For this reason, state estimation with the EnKF leveraging instantaneous turbine power measure-
10 ments, using an upstream-pointing LiDAR, and using a downstream-pointing LiDAR are compared in Fig. 6. Flow and power measurements are artificially disturbed by zero-mean white Gaussian noise with $\sigma = 0.10$ m/s and $\sigma = 10^4$ W, respectively.





**Figure 5.** Comparison of estimation errors in flow fields for state-only estimation with the ExKF, EnKF and UKF at $t = 300$ s and $t = 700$ s. The model and KF settings are depicted in Tables 1, 2, and 3. Wind is coming in from the top, flowing towards the bottom. The KFs consistently improve the instantaneous flow field estimations, noticeably near the turbine rotors, where measurements (red dots) are nearby.

The KF settings are displayed in Tables 2 and 3. In Fig. 6 it can be seen that SCADA data allows comparable performance compared to the use of flow measurements, making the proposed closed-loop control solution feasible for implementation in



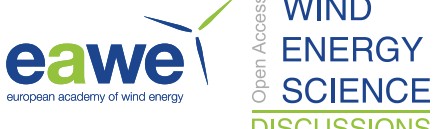

**Figure 6.** Comparison of estimation errors in flow fields for state-only estimation with the EnKF for various sensor configurations: using only power measurements (SCADA), using flow measurements with a LiDAR system pointing upstream, and using flow measurements with a LiDAR system pointing downstream of the rotor. The freestream wind is coming in from the top of the page, and flows towards the bottom. Measurements (turbine/flow dots) are indicated in red.

existing wind farms, without the need for additional equipment. Furthermore, this modular framework allows the use of a combination of LiDAR systems, measurement towers, and/or SCADA data, whichever is available, for model calibration.



### 4.2.3 State and parameter (dual) estimation

Accurate long-term forecasting demands the calibration of model parameters such as $\ell_s$ in addition to the states (flow fields). Dual estimation using flow measurements downstream of each turbine disturbed by zero-mean white noise with $\sigma = 0.10$ m/s (as shown in the rightmost plots in Fig. 6) for the EnKF and UKF is displayed in Fig. 7, where the turbulence model tuning

parameter $\ell_s$ is additionally estimated. The higher $\ell_s$, the more wake recovery is modeled by WFSim. The KF settings are identical to those in Tables 2 and 3. From this figure, it becomes clear that the flow field estimates are not only improved for short-term forecasting, but are also consistently better than the non-calibrated (open-loop) model's forecast for longer-term forecasting due to the real-time adaptation of the turbulence model to the actual atmospheric conditions. Furthermore, it can be seen that the EnKF performs comparably to the UKF, but at a much lower computational cost. Note that the EnKF even

outperforms the UKF in this simulation, but this is expected to be due to randomness in the EnKF. On average, the EnKF is expected to perform similar to the UKF in terms of estimation accuracy.

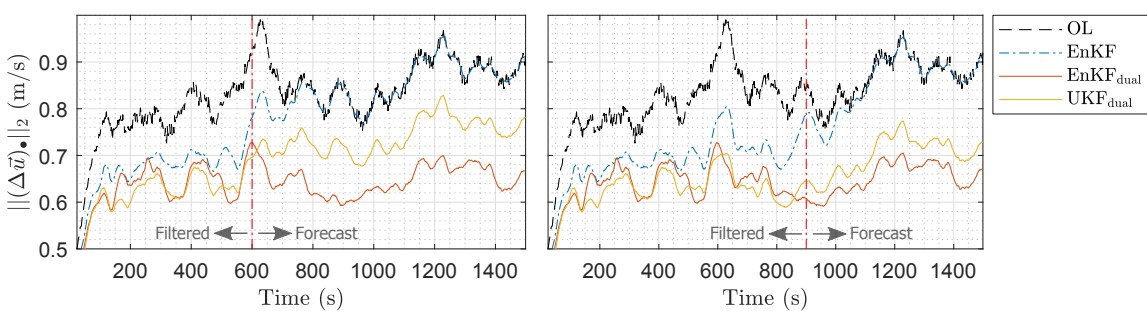

**Figure 7.** Comparison of forecasting performance for state-only and state-parameter ($\ell_s$) dual estimation with the EnKF and UKF, where measurements are available up until the vertical red dashed lines, after which the estimation becomes a forecast. The KFs consistently improve both the short-term and long-term forecasts, since the turbulence model is now also calibrated.

### 4.3 9-turbine ALM with turbulent inflow

In this section, we investigate the performance of the model calibration solution under a realistic 9-turbine wind farm scenario, which employs the EnKF for dual state-parameter estimation. The wind farm contains nine NREL 5-MW baseline turbines,

oriented in a three by three layout, separated five and three rotor diameters apart in streamwise and crosswise direction, respectively. The turbines start with a $30°$ yaw misalignment, but are aligned with the mean wind direction within the first 30 s of simulation. The turbine layout and numbering is shown in the top-left subplot of Fig. 9. This LES simulation has been used before in the literature, and is described in more detail in Boersma et al. (2017b). A number of important simulation properties are listed in Table 4 for SOWFA and WFSim, respectively.

Note that $N$ has increased by a factor $4$. In the UKF, this would result in the same factor of additional particles. Thus, not only is each particle more expensive to calculate, there are also more particles. Rather, in the EnKF, the approach is heuristic. None of the EnKF settings needed to be changed for good performance compared to Section 4.2, as displayed in Tables 2 and 3.





**Table 4.** Overview of several settings for the SOWFA and the WFSim 9-turbine wind farm simulation.

| Variable | Symbol | SOWFA | WFSim |
|---|---|---|---|
| Domain size | - | $3.5\text{km} \times 3.0\text{km} \times 1.0\text{km}$ | $1.9\text{km} \times 0.80\text{km}$ |
| Number of states | $N$ | $\mathcal{O}(10^8) - \mathcal{O}(10^9)$ | $1.2 \cdot 10^4$ |
| Cell size near rotors | - | $3\text{m} \times 3\text{m} \times 3\text{m}$ | $25\text{m} \times 38\text{m}$ |
| Cell size outer regions | - | $12\text{m} \times 12\text{m} \times 12\text{m}$ | $25\text{m} \times 38\text{m}$ |
| Rotor model | - | ALM | ADM ($c_f = 2.0$, $c_p = 0.97$) |
| Inflow wind speed | $U_\infty$ | 12.03 m/s | 12.00 m/s (OL)<br>9.00 m/s (EnKF) |
| Atmospheric turbulence | - | $\text{TI}_\infty = 4.7\%$ | $d' = 3.8 \cdot 10^1$ m<br>$d = 5.2 \cdot 10^2$ m<br>$\ell_s = 3.9 \cdot 10^{-2}$ |

As shown in Table 3, the EnKF has a low computational cost of 1.2 s/iteration (8 cores, parallel). In this case study, both the complete model state (flow field), the turbulence model parameter $\ell_s$, and the freestream flow speed $U_\infty$ are estimated in real-time using exclusively readily available power measurements from the turbines. The EnKF will deliberately be initialized with a poor value for $\ell_s$ and $U_\infty$ to investigate convergence. The performance will be compared to an open-loop simulation of

5 WFSim with a poor value for $\ell_s$, but with a correct value for $U_\infty$.

In Fig. 8, it can be seen that the EnKF is successful in estimating the freestream wind speed $U_\infty$ and the turbulence model parameter $\ell_s$ after about 300 s using only wind turbine power measurements. Furthermore, the flow fields of the to-be-estimated model (SOWFA), of the open-loop (OL) simulation, and of the EnKF at various time instants are displayed in Fig. 9. From this figure, it can be seen that the EnKF has very large errors at the start of the simulation. However, after 10 s, the error in flow

10 states surrounding each turbine significantly decreases through the use of wind turbine power measurements. This estimated flow then propagates downstream, "clearing up" the errors in the vicinity of the wind turbines. As time further propagates, the freestream estimation improves, and the errors in front of the first row of turbines also reduce. Finally, the turbulence model also adapts and the EnKF outperforms the open-loop simulation consistently.

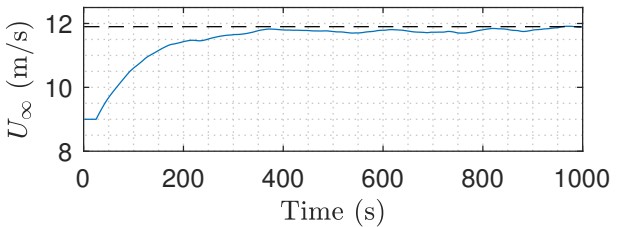
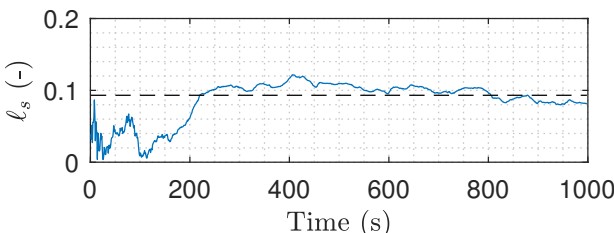

**Figure 8.** Convergence of $\ell_s$ and $U_\infty$ using the EnKF. In dashed lines are the grid-searched optimal constant values for the open-loop simulation. With power measurements only, the model is able to estimate these parameters successfully in addition to the model states.





**Figure 9.** Comparison of flow fields for state-parameter estimation with the EnKF. Wind is coming in from the top and flows downwards. The variables $U_\infty$ and $\ell_s$ are incorrectly initialized in the EnKF and estimated in addition to the states, using only turbine power measurements. The open-loop (OL) simulation is initialized with a poor $\ell_s$ but correct $U_\infty$. The EnKF quickly converges for the states, and more slowly for $\ell_s$ and $U_\infty$. After several hundreds of seconds, the EnKF has converged and consistently reconstructs the wind flow in the farm.



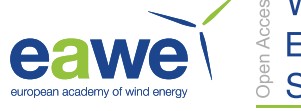

**Figure 10.** Comparison of power forecasting using the EnKF with measurements available up until time $t = 600$ s. After convergence of the freestream wind speed (as seen as a positive power slope for the first row of turbines), the turbulence model is also calibrated. After convergence, forecasting is significantly better than in open-loop. Oscillatory behavior is still present due to an oscillatory input signal ($C_T'$), turbulent flow field, and the absence of inertia in the rotor model. Adding rotor inertia in the surrogate model would smooth the results to better resemble true power data.

The power forecasting performance is shown in Fig. 10. In this figure, the power forecast for the OL is compared to that of the EnKF, where we define the error in the time-series of the generated power of a single turbine $i$ as $(\Delta P)_\bullet^i \in \mathbb{R}^{T_k - T_f}$ as

$$(\Delta P)_\bullet^i = \left[ P_{k=T_f+1}^i - P_{\text{SOWFA},k=T_f+1}^i \quad P_{k=T_f+2}^i - P_{\text{SOWFA},k=T_f+2}^i \quad \cdots \quad P_{k=T_k}^i - P_{\text{SOWFA},k=T_k}^i \right]^T,$$

with $T_k$ the total number of discrete simulation timesteps, and $T_f$ the discrete timestep at which the forecast starts.



As previously seen in Fig. 8, the EnKF converges reasonably well after 300 s, and indeed the power forecasts outperform those of the OL system at $t = 300$ s. Furthermore, it is interesting to see that the filtered power estimates of the first row of turbines ($i = 1, 2, 3$) starts low at $t = 1$ s, but converges to the true power at approximately $t = 200$ s. This can be related to the mismatch in $U_\infty$ estimates, which takes approximately $200 - 400$ s to converge to the true value of 12 m/s, as seen in Fig. 8.

The oscillatory behavior in both the OL and EnKF power predictions is due to the absence of rotor inertia in the rotor model, turbulent structures in the flow, and large fluctuations on the excitation signal $C_T'$.

Finally, the forecasts for flow at times $t = 300$ s and $t = 600$ s are examined in Fig. 11. The large flow estimation mismatch in the EnKF at $t < 250$ s quickly reduces and for $t > 250$ s the EnKF estimation is consistently better than the OL case. This has to do with the convergence of the model parameters $\ell_s$ and $U_\infty$, and the estimation of the states surrounding the turbines

using the power measurements.

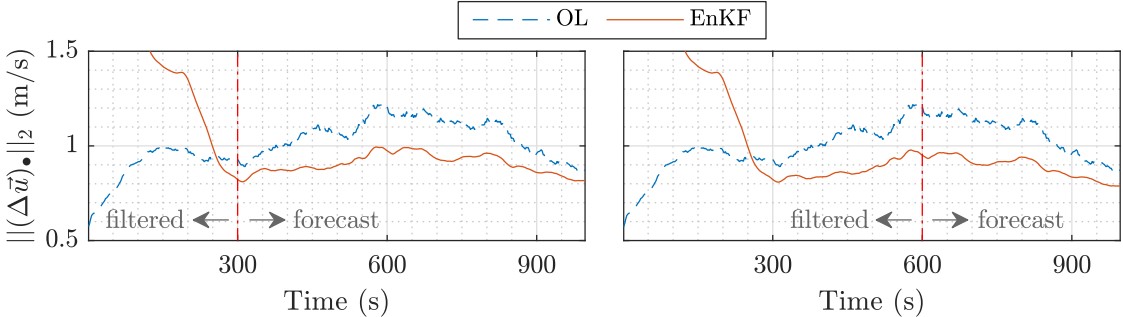

**Figure 11.** Comparison of flow field estimation using the EnKF. On the left, measurements are available only until $t = 300$ s, and on the right until $t = 600$ s. The EnKF first has to converge due to the large mismatch between the estimated and true $U_\infty$. After convergence, the forecasts are significantly better than in open-loop. Note that a relatively large region of (poorly observable) freestream flow is included in $(\Delta \boldsymbol{u})$, and hence the results appear suppressed compared to Fig. 9.

## 5   Conclusion

This paper presented a real-time model calibration algorithm for the dynamic surrogate wind farm model "WFSim", relying on an Ensemble Kalman filter (EnKF) at its core. The calibration solution was tested in two distinct high-fidelity wind farm simulations. Using exclusively SCADA measurements which are readily available in current wind farms, the adaptability to

model discrepancies and time-varying atmospheric conditions (namely, the turbulence intensity and freestream wind speed) in a 9-turbine wind farm simulation was shown, at a low computational cost of approximately 1 s per timestep. Specifically, the atmospheric parameters were shown to converge to their optimal values within 300 s. Furthermore, the EnKF was shown to perform comparably in terms of accuracy to the state-of-the-art algorithms in the literature, at a computational cost of multiple orders of magnitude lower. Additionally, estimation using flow measurements from LiDAR was compared to estimation using

SCADA data, and it was shown that SCADA data can effectively be used for real-time model calibration. Using the proposed




adaptation solution, the calibrated wind farm model can be used for accurate forecasting and optimization. This work presented an essential building block for closed-loop wind farm control using surrogate dynamic wind farm models.

*Code and data availability.*

All models and algorithms presented in this article are open-source. The surrogate wind farm model (WFSim) and the
5  calibration solution are available in the public domain at https://github.com/TUDelft-DataDrivenControl/. SOWFA is available at https://github.com/NREL/SOWFA. All rights for SOWFA belong to the National Renewable Energy Laboratory (NREL) for performing the simulations and providing these data.

*Acknowledgements.* The authors would like to thank Matti Morzfeld for the insightful discussions concerning the Ensemble Kalman filter. However, any mistakes in this work remain the authors' own. This project has received funding from the European Union's Horizon 2020
10  research and innovation program under grant agreement No 727477. 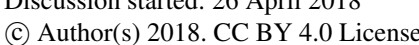




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
