# Peer review of "Online model calibration for a simplified LES model in pursuit of real-time closed-loop wind farm control"

_Wind Energy Science, 2018_

## Referee Comment (RC1) · Anonymous Referee #1 · 4 May 2018

General comments:

Good and relevant work.

Specific comments:

I have concerns about the validity of neglecting the vertical dimension, given significant effects such as vertical meandering, wind shear and veer. There should be more comment about this.

In the introduction, some re-wording is necessary: "all commercial turbines are currently disconnected from the electricity grid by their power electronics" is not right. They

are certainly connected, it's only that the rotational speed is decoupled from the grid frequency. Also in the next line, "grid-disconnected renewable energy plants": maybe "non-synchronous" or some other expression, but not "grid-disconnected"!

Technical corrections:

Equation (5): Actually there is now plenty of evidence that the cos-cubed law for the effect of yaw on power coefficient is not at all correct. There should at least be some comment about this.

---

## Referee Comment (RC2) · Anonymous Referee #2 · 5 Jun 2018

This paper is about state estimation for a simplified CFD model, aiming to be used in a feedback control setting.

General comments: The paper is interesting, but way too long compared to its actual contribution. I suggest it is reconsidered for publication after major review.

Major comments:

1. My main comment is: reduce the length substantially to increase the readability, but most importantly, to clarify the actual contribution of the paper. Remove details that are previously published. More specifically,

- in the introduction, focus on the actual innovation of the work and its relation to other work.

- The detailed explanation of the surrogate model (Section 2) can be removed, providing the proper reference to the original work. Of course, a shorter summary of the model will be welcome

- Details about the different forms of (standard) Kalman filters are also not necessary, a reference is sufficient, like linear and extended KF.

2. Section 4 but wel written and clear, but I recommend the following changes:

- Sec 4.2.1 The improvement with respect to the estimation-free case does not seem to be very pronounced. Can you comment more on that. The state estimation adds significant computational complexity, which requires more significant benefits in terms of accuracy. Maybe a different simulation scenario, including changes in the wind conditions during the simulaton, can be used to better demonstrate the performance.

- Sec 4.2.2, Fig 6: please show the model output (WT power) estimation and compare to the true output. At both turbines the wind speed is estimated clearly higher than the simulated one, implying that the estimated power will also be (much!) higher.

- Fig 7: it is essential here to simulate using a realistic low frequency variations in the wind condition. Obviously, the prediction will be good when the incoming wind speed and direction does not change, since the underlying model does not change. The different KF's outperform the OL simulation in the prediction ONLY because the initial condition at the beginning of the prediction part is worse for the OL case (the KF based models have been adapted during the estimation phase, and the OL model has not). I suggest you focus in Figure 7 on the estimation part and remove the prediction part. Same holds for Figure 10.

- Sec 4.3: the performance is compared to the open-loop simulation with the correct wind velocity U_inf, while the EnKF is initialized with a different U_inf. For better comparison I recommend you compare to the OL simulation with the same initial condition. That will not only be fair, but actually also in favor of your approach.

- Fig 11. It would make more sense to focus the comparison only on the wind velocity in the wakes, rather than the whole wind field.

3. It is claied on several occations in the paper that the presented estimator is useful for performing long-term forcasting. I disagree with this statement, and if you want to convince me then you would have to demonstrate the ability of the estimator to make correct predictions in the future when the input conditions (eg ambient wind velocity) vary in time. For instance, to predict upcoming wind speed or direction changes. Obviously, this will not be possible, so I suggest that with respect to the application of the estimator you stick to feedback control. When used for MPC, I suggest using the term short-term prediction.

Minor comments:

- p.2, line 9 - remove bracket

- p.6, line 2 - abbrev ADM seems not used in the sequel, check and if so - remove. Same holds for the abbrev. UT on page 11, line 15

- p.6, eq. 4: define $\phi$, D and $C_{T_i}\hat{}$' (what do you mean by variation?). Also, does the term H[] not imply that the thrust force is excerted within a circle around the turbine, rather than on the rotor plane/line?

- p. 8, bottom: notation already defined in Sec 2.4

- p. 9 bottom: why is $P_{k|k-1}\hat{}z$ is not necessarily invertible, then $L_k$ will not be full rank (or possibly even $Lk=0$), implying that some (or all) measurements will not contribute to the state estimation

- p.11, eq. (21): define all used notation

- p.13, eq (27) : how are the estimates $\hat w_{k-1}\hat{}i$ and $\hat w_{k}\hat{}i$ obtained and

updated?

- p 16, line 1: are you suggesting to use the wind vane measurements of only the upstream turbines to estimate \phi, or all turbines? Please be clear, because if you use all turbines you will be neglecting the dynamics of the propagation of the wind direction through the wind farm.

- p.18, line 14: estimation -> you mean "parameter estimation"?

- p.21 Fig 6 caption: "The freestream wind is coming in from the top of the page, and flows towards the bottom." Isn't it the other way around?

- p.22 line 3: "Dual estimation using flow measurements downstream..." - please explain the used measurements in more detail

- p22, line 4: exaplin figure 7 clearly. What's on the y axis?

- page 25, equation on bottom: bullet notation not clear

---

## Author Comment (AC1) · 5 Jul 2018

Dear Reviewers,

The authors express their gratitude to the reviewers for the time and effort spent to provide positive and constructive feedback to the submitted WES manuscript. Their comments will play a crucial role in further improving the scientific quality and relevance of this work. In accordance to the provided feedback, the article will be revised appropriately. The objective of the attached document is to respond to all the concerns raised by each reviewer, and explain how the authors will address each issue in the revised article.

Yours sincerely,

Bart Doekemeijer

Please also note the supplement to this comment:
https://www.wind-energ-sci-discuss.net/wes-2018-33/wes-2018-33-AC1-supplement.pdf

**Supplement:**

| | |
|---|---|
| Date | July 5, 2018 |
| Our reference | WES-2018-33 |
| Contact person | B M Doekemeijer |
| Telephone | +31 15 278 5623 |
| E-mail | b.m.doekemeijer@tudelft.nl |
| Subject | Response to reviewers |

**Delft University of Technology**

Delft Center for Systems and Control

Address
Mekelweg 2 (3ME building)
2628 CD Delft
The Netherlands

www.dcsc.tudelft.nl

Reviewers
*Wind Energy Science Discussions*

Dear Reviewers,

The authors express their gratitude to the reviewers for the time and effort spent to provide positive and constructive feedback to the submitted WES manuscript. Their comments will play a crucial role in further improving the scientific quality and relevance of this work. In accordance to the provided feedback, the article will be revised appropriately. The objective of the attached document is to respond to all the concerns raised by each reviewer, and explain how the authors will address each issue in the revised article.

Yours sincerely,

Bart Doekemeijer

Enclosure(s): Response to comments of Reviewer 1
Response to comments of Reviewer 2

Date    July 5, 2018
Reference    WES-2018-33

**Response to comments of Reviewer 1**

**General comments**

- Good and relevant work.
  We thank and appreciate the positive feedback of the reviewer.

**Specific comments**

- **Feedback**: I have concerns about the validity of neglecting the vertical dimension, given significant effects such as vertical meandering, wind shear and veer. There should be more comment about this.

  **Response**: The reviewer raises an important issue: the validity of the control-oriented model. This has also been a concern of the authors in previous work, and has been tackled in the original article of Boersma et al. [2017]. That paper presents and discusses the control-oriented model "WFSim" in greater detail, including the complete mathematical derivation and comparisons against two different high-fidelity large-eddy simulation codes. In the current WES submission, the aim is to provide a concise yet informative overview of the mathematical model used in the remainder of the article.

  Furthermore, from an estimation point of view, indeed the model should be as accurate as possible. However, from a controls perspective, the computational cost limits us to models that are fast enough for real-time application, and therefore the accuracy is limited. The presented estimation algorithm has the potential to account for modeling errors (e.g., vertical meandering, wind shear, and wind veer, as the reviewer rightfully mentioned) by assimilating real-time measurements into the simplified mathematical model. In addition to the simulation results presented in the work at hand, the simulation results presented in the work of Boersma et al. [2017] show that the model matches well in terms of the hub-height flow and the turbine power production compared to a high-fidelity large-eddy simulation code, which *does* include the aforementioned atmospheric effects. While these simulations are not conclusive, it at least provides a safe ground to state that the control-oriented model is valid for the demonstrated cases.

  **Revision plan**: The reviewer raises an important concern, and it becomes apparent to the authors that they should clearly define the scope of this paper, and the scope of each section. Furthermore, the assumptions in the mathematical model will be motivated more explicitly, and the reader will be referred to Boersma et al. [2017] for more information.

- **Feedback**: In the introduction, some re-wording is necessary: "all commercial turbines ... power electronics" is not right. They are certainly connected, it's only that the rotational speed is decoupled from the grid frequency. Also, in the next line, "grid-disconnected renewable energy plants": maybe "non-synchronous" or some other expression, but not "grid-disconnected"!

  **Response**: The reviewer is correct, and the proposed changes will be made to the introductory text.

  **Revision plan**: The text will be revised in accordance to the suggestions.
    - Page 1. "*While there are ... the rotational speed of almost all commercial turbines is currently decoupled from the electricity grid frequency via each turbine's power electronics (Aho et al., 2012).* "
    - Page 1. "*As the current grid-connected ... replaced by non-synchronous renewable energy plants, the inertia of the electricity grid will decrease.*"

**Technical corrections**

- **Feedback**: Equation (5): Actually there is now plenty of evidence that the cos-cubed law for the effect of yaw on power coefficient is not at all correct. There should at least be some comment about this.

  **Response**: The authors thank the reviewer for raising this concern. This issue is in line with the comment concerning model validity. The reviewer is correct that other control-oriented models such as FLORIS [Gebraad et al., 2016] use a different correction term to incorporate a yaw misalignment in the turbine power expression. Unfortunately, to the best of the authors' knowledge, no systematic way is described in the literature on how to best choose this exponential term, besides through a comparison with high-fidelity data, as also discussed in the original paper by Boersma et al. [2017]. In the presented "WFSim" model, the cosine-cubed law originates from a physical derivation of the Navier-Stokes equations where the turbines are modeled using actuator disk theory. It might indeed turn out that, for a certain topology and turbine type, the cubed-law will not suffice, and a different exponent should be assumed. The effect of this discrepancy does not further impinge on the results presented in this article, as all simulations are with non-yawed turbines.

  **Revision plan**: In accordance with the comment of the author, the assumptions and limitations underlying the cosine-cubed law in Equation (5) will be described, including an explicit reference to the original paper on WFSim by Boersma et al. [2017]. The original paper gives a more thorough motivation for this power law. As the aim of the current WES manuscript is not to introduce, define, nor validate but rather to summarize the model, the authors hope that the reviewer finds this sufficient.

**Response to comments of Reviewer 2**

**General comments**

- **Feedback**: The paper is interesting, but way too long compared to its actual contribution. I suggest it is reconsidered for publication after major review.
  **Response**: We thank and appreciate the constructive feedback of the reviewer. The authors understand the concern of the reviewer, and have also had internal discussions before submission on whether or not to include certain sections. The purpose of this paper is to present a complete and unifying framework for joint state-parameter estimation for a control-oriented wind farm model, and therefore it was decided to include certain levels of detail. A more explicit motivation is found in the following responses.

**Major comments**

1. **Feedback**: My main comment is: reduce the length substantially to increase the readability, but most importantly, to clarify the actual contribution of the paper. Remove details that are previously published. More specifically,

   - in the introduction, focus on the actual innovation of the work and its relation to other work.
     **Response**: Currently, the complete closed-loop wind farm control framework is presented in Fig. 1, including the place of this work's contribution in relation to the other components. Furthermore, this (closed-loop) framework is compared against other (open-loop & model-free) frameworks on page 2. For each component of the presented framework, the state of the art is presented and compared to the work of the authors. The actual innovation of this work is described on page 3, where the current contributions compared to previous work are detailed.

- The detailed explanation of the surrogate model (Section 2) can be removed, providing the proper reference to the original work. Of course, a shorter summary of the model will be welcome.

  **Response**: The authors understand the concern of the reviewer. The authors had considered the removal of sections 2 and 3 before submission, but it was decided not to, because the purpose of this article is to present a unifying framework, including enough detail for unfamiliar readers (without the necessary aerodynamics and/or state estimation background) to understand the work to a sufficient degree. The model is presented in such detail to motivate the choice and understanding of the states and parameters that are to be estimated, and to help interpret the results found in section 4. For example, the mixing length parameter $\ell_s$ is estimated in certain simulations in section 4, and therefore the turbulence model is presented beforehand in section 2. Furthermore, the mathematical notation of section 2 is in agreement with that of sections 3 and 4, making it easier for unfamiliar readers to follow.

  **Revision plan**: The authors will revise the entire document to address the issue of the paper's length, aiming to reduce the number of words by at least 10 %.

- Details about the different forms of (standard) Kalman filters are also not necessary, a reference is sufficient, like linear and extended KF.

  **Response**: We appreciate the reviewer's constructive feedback. This concern is in line with the previous comment concerning section 2. It is apparent that the reviewer is an expert in the field of control and state estimation. However, the target audience for this article is not exclusively control engineers, but to the wider range of the wind energy community. The purpose of this article is to present a unifying framework. The various Kalman filters are mathematically introduced to ensure reproducibility and to promote clarity and understanding when interpreting the results presented in section 4. The mathematical model formulation, the Kalman filtering section, and the results all maintain a homogeneous symbol notation and follow one another logically, providing clarity in the remainder of this article.

  **Revision plan**: The authors will revise the entire document to address the issue of the paper's length, aiming to reduce the number of words by at least 10 %.

2. **Feedback**: Section 4 but wel written and clear, ...

**Response**: We appreciate the positive feedback on section 4. The authors also believe that this clarity is largely due to the unifying framework presented, which led to an article that is longer than average.

...but I recommend the following changes:

- **Feedback**: Sec 4.2.1 The improvement with respect to the estimation-free case does not seem to be very pronounced. Can you comment more on that. The state estimation adds significant computational complexity, which requires more significant benefits in terms of accuracy. Maybe a different simulation scenario, including changes in the wind conditions during the simulaton, can be used to better demonstrate the performance.

  **Response**: The reviewer addresses a very important topic: the usefulness of the estimator compared to the addition in computational cost. In Figure 5, an open-loop simulation ("WFSim") is compared to closed-loop simulations in which the states of WFSim are estimated using a variety of Kalman filters. For the 2-turbine simulation, the WFSim model is accurate for the first wake, and there is a negligible increase in accuracy. For the second wake, the improvements are more noticeable, and the presented simulation case suffices for the comparison of the different Kalman filter algorithms. The reviewer is correct to state that the case does not necessarily motivate the use of state estimation in the 2-turbine case. However, the 2-turbine simulation study showcases the differences in Kalman filters and sensor locations, building up towards the more realistic 9-turbine wind farm simulation. In the 9-turbine simulation study, the need for state estimation becomes more apparent.

  **Revision plan**: The authors will more carefully highlight the purpose of the 2-turbine case study and the 9-turbine case study.

- **Feedback**: Sec 4.2.2, Fig 6: please show the model output (WT power) estimation and compare to the true output. At both turbines the wind speed is estimated clearly higher than the simulated one, implying that the estimated power will also be (much!) higher.

  **Response**: The authors appreciate the reviewer's idea of showing the turbine power signals, be it that it will elongate the article even further. However, the authors are confused about the statement that the estimated turbine power is higher than the simulated power. Figure 6 shows the absolute value of the *error* in the flow velocities. For clarification, the power signals are plotted next of 1) the simulation where the EnKF uses power measurements, 2) the simulation in which upstream flow measurements are used, and 3) the simulation in which downstream flow measurements are used:

[Figure]

**Power measurements**

[Figure]

**Upwind lidar measurements**

[Figure]

**Downwind lidar measurements**

From these figures, the difference between true and estimated power is small. Note that the red dots and red turbines in Fig. 6 of the paper are indicating that they use flow and turbine power measurements, respectively, rather than their error value. This may have caused confusion. The inflow at each turbine shows a close-to-blue color, indicating that the estimation error in wind speed is very small, which fits with the small error in power estimation from the figures above.

**Revision plan**: The authors will more carefully emphasize that what is being plotted in Figures 5 and 6 and the right of Figure 9 are absolute values of the errors.

- **Feedback**: Fig 7: it is essential here to simulate using a realistic low frequency variations in the wind condition. Obviously, the prediction will be good when the incoming wind speed and direction does not change, since the underlying model does not change. The different KF's outperform the OL simulation in the prediction ONLY because the initial condition at the beginning of the prediction part is worse for the OL case (the KF based models have been adapted during the estimation phase, and the OL model has not). I suggest you focus in Figure 7 on the estimation part and remove the prediction part. Same holds for Figure 10.

**Response**: The authors thank the reviewer for the valuable comment and suggestions on further simulation studies. The authors think that it is not obvious that the predictions in Figure 7 will be accurate. Note that only two measurements are available at each timestep, while 1 model parameter and 3200 states are estimated simultaneously. As observability is not a given, the model is nonlinear and can go unstable, the turbines are following a quickly varying excitation signal, and we are feeding in noisy measurements of a significantly different, high-fidelity model, the authors believe it is not a given that the prediction will be accurate.

Furthermore, the purpose of showing the prediction is to highlight the difference in state-only against state & parameter estimation. In state-only estimation, indeed only the initial condition is better than the OL case. The state-only estimation forecast will converge to the OL forecast after a certain amount of time (around $t = 800$ s and $1100$ s, respectively). However, for the state & parameter estimation case, the estimate should outperform the state-only estimation for larger prediction horizons, showing the importance of parameter estimation in addition to state estimation.

Further, the authors think it is a great suggestion to investigate the performance under a realistic, time-varying and spatially-varying inflow. Currently, the turbulent inflow for SOWFA is generated following a precursor simulation, in which a realistic turbulent flow field is developed. As the reviewer rightfully mentions, while there are turbulent fluctuations in the inflow, this is with a single *mean* wind speed and wind direction. However, to generate a realistic inflow with low-frequency changes in the ambient conditions for a high-fidelity wind farm simulation, one would need to couple a mesoscale model with a large-eddy simulation model. This is a scientific study by itself (e.g., Rodrigo et al. [2016], Santoni et al. [2018]), and considered outside of the scope of this work.

The authors still believe there is value in the work presented in this article. Currently, the solution has been tested in a high-fidelity simulation environment with a realistic, turbulent inflow. Furthermore, convergence of the wind speed is shown, by purposely initializing the model internal to the estimation algorithm with a wrong freestream wind speed, with success. This suggests that in the comparable situation that when the actual freestream wind speed changes, the solution will succeed too.

An alternative to high-fidelity simulation would be to use experimental data. However, such data is currently difficult to obtain for the authors. Furthermore, it is questionable whether a wind tunnel provides an inflow with realistic low-frequency variations in ambient conditions, and flow scaling may become a problem too. Experimental full-scale data would be more sensible, but this will yield other issues such as confidentiality, the scientific relevance of the turbines in the farm, signal synchronization, sensor uncertainties and inconsistencies, and the lack of (reliable) flow and turbine measurements.

**Revision plan**: The need for high-fidelity data with realistic inflow conditions, including low-frequency changes in the wind direction, turbulence intensity, and wind speed will be presented as an important next step for future work, and the limitations of the current simulations will be described more explicitly.

- **Feedback**: Sec 4.3: the performance is compared to the open-loop simulation with the correct wind velocity U_inf, while the EnKF is initialized with a different U_inf. For better comparison I recommend you compare to the OL simulation with the same initial condition. That will not only be fair, but actually also in favor of your approach.

  **Response**: The reviewer has a good point: for a fair comparison, the authors should "compare apples to apples": use the same initial conditions in the OL simulation as in the KF simulation.

  **Revision plan**: The authors will redo the OL simulations with a freestream wind speed of $9$ m/s, and update the results presented in Section 4.3.

- **Feedback**: Fig 11. It would make more sense to focus the comparison only on the wind velocity in the wakes, rather than the whole wind field.

  **Response**: The reviewer makes a good point concerning the region of interest when comparing flow fields, and this has been a concern of the authors too in the past. However, the authors believe that it may also be of interest to accurately estimate the non-waked flow surrounding the wind turbines, as this may influence the optimal control strategy in the closed-loop framework later on. Namely, an optimal yaw steering strategy may depend on the non-waked flow conditions on either side of the farm. Furthermore, the definition of a waked region and the extraction of the flow velocities therein requires additional work and explanation, while the dominant trends are identical when considering the complete flow field. Note that the inclusion of the unwaked flow in the surrogate model is important for stability and to reduce boundary effects of the model.

3. **Feedback**: It is claimed on several occations in the paper that the presented estimator is useful for performing long-term forcasting. I disagree with this statement, and if you want to convince me then you would have to demonstrate the ability of the estimator to make correct predictions in the future when the input conditions (eg ambient wind velocity) vary in time. For instance, to predict upcoming wind speed or direction changes. Obviously, this will not be possible, so I suggest that with respect to the application of the estimator you stick to feedback control. When used for MPC, I suggest using the term short-term prediction.

   **Response**: The reviewer makes a very good point, and there is not sufficient evidence to claim that our proposed solution can consistently and reliably provide long-term forecasting. Furthermore, after revision, the authors agree that the definition "long-term" has not been defined appropriately in the article. While the idea should not be discarded, there is not enough evidence supporting the claim as of right now. Further, the high-fidelity simulation with realistic low-frequency changes in the ambient conditions is outside of the scope of this article.

   **Revision plan**: As suggested, the claims concerning long-term forecasting will be removed. The claims for forecasting, including short-term forecasting, will be rephrased to address the limitations. Namely, time-varying wind directions and wind speeds have not been considered.

**Minor comments**

- **Feedback**: p.2, line 9 - remove bracket

  **Response**: The authors express gratitude for the time and effort invested by the reviewer to read the document so carefully.

  **Revision plan**: The suggested changes will be addressed in a revised version.

- **Feedback**: p.6, line 2 - abbrev ADM seems not used in the sequel, check and if so - remove. Same holds for the abbrev. UT on page 11, line 15.

  **Revision plan**: The suggested changes will be addressed in a revised version.

- **Feedback**: p.6, eq. 4: define $\phi$, D and $C'_{T_i}$ (what do you mean by variation?). Also, does the term H[] not imply that the thrust force is excerted within a circle around the turbine, rather than on the rotor plane/line?

  **Response**: With *variation*, a parametrization is meant. There is a one-to-one mapping between the traditional thrust coefficient and $C'_T$. The latter has been used more popularly in the work by Goit and Meyers [2015] and Munters and Meyers [2017], and this is the way it has been defined in the original paper by Boersma et al. [2017]. Furthermore, the reviewer is correct that $H[\bullet]$ implies a circle around the turbine center. However, the additional term $\delta[\bullet]$ projects this circle onto the rotor plane, leading to the actuator disk implementation.

  **Revision plan**: The symbols will be defined as suggested. Furthermore, Equation (4) will be explained more carefully to avoid confusion.

- **Feedback**: p. 8, bottom: notation already defined in Sec 2.4

  **Response**: The authors appreciate the reviewer's eye for detail.

  **Revision plan**: The repeated definition of symbols will be omitted in the revised version.

- **Feedback**: p. 9 bottom: why is $P^z_{k|k-1}$ is not necessarily invertible, then $L_k$ will not be full rank (or possibly even $L_k = 0$), implying that some (or all) measurements will not contribute to the state estimation

  **Response**: The reviewer makes a justified comment concerning the invertibility of the covariance matrix, and the state updates that it results in. The issue of invertibility is closely related to how the covariance matrices are calculated. Specifically, for the sample-based algorithms, it may occur that singular covariance matrices arise (see Equation (28)). This is especially the case when the number of samples is smaller than the number of system states. As the reviewer rightfully mentions, this may result into a Kalman gain $L_k$ which is not full rank. In that situation, certain measurements may not be used to update the state vector. Since the samples in the Ensemble KF and thus the rank properties of the covariance matrix change in each timestep due to the random Gaussian noise, this is not expected to be an issue.

- **Feedback**: p.11, eq. (21): define all used notation

  **Revision plan**: The definition of $\overline{\psi}_{k-1|k-1}$ in Equation (21) will be introduced in the text, and the definition of $N$ will be repeated for clarity.

- **Feedback**: p.13, eq (27) : how are the estimates $\hat{\boldsymbol{w}}_{k-1}^i$ and $\hat{\boldsymbol{w}}_k^i$ obtained and updated?

  **Response**: The variables $\hat{\boldsymbol{w}}_{k-1}^i$ and $\hat{\boldsymbol{v}}_k^i$ are realizations of zero-mean Gaussian white noise, where the covariance is defined through Equation (8), which reads:

  $$\mathrm{E}\left[\begin{bmatrix}\boldsymbol{v}_k \\ \boldsymbol{w}_k\end{bmatrix}\begin{bmatrix}\boldsymbol{v}_\ell^T & \boldsymbol{w}_\ell^T\end{bmatrix}\right] = \begin{bmatrix}\boldsymbol{R}_k & \boldsymbol{S}_k^T \\ \boldsymbol{S}_k & \boldsymbol{Q}_k\end{bmatrix}\Delta_{k-\ell}, \quad \text{where} \quad \Delta_{k-\ell} = \begin{cases}1, & \text{if } k = \ell, \\ 0, & \text{otherwise.}\end{cases}$$

  In practice, these noise terms are generated using MATLAB's `randn()` command, employing a constant preloaded random seed between simulations for one-to-one comparisons between different Kalman filtering algorithms.

  **Revision plan**: A more explicit explanation will be introduced near Equation (27), detailing how $\hat{\boldsymbol{w}}_{k-1}^i$ and $\hat{\boldsymbol{v}}_k^i$ are calculated.

- **Feedback**: p 16, line 1: are you suggesting to use the wind vane measurements of only the upstream turbines to estimate $\phi$, or all turbines? Please be clear, because if you use all turbines you will be neglecting the dynamics of the propagation of the wind direction through the wind farm.

  **Response**: The reviewer raises an important question concerning the determination of the wind direction, $\phi$. Currently, the wind direction is calculated as the average of the wind vane measurements of *all* turbines inside the farm, both up- and downstream. The issue that the reviewer mentions has not explicitly been considered in this work. Based on the available data from high-fidelity large-eddy simulations, no significant differences were found between the wind vane measurements for the various wind turbines. The main motivation for the use of all the turbine vane measurements was to reduce the variance by using all measurements as they are assumed to measure the same thing with (more or less) independent errors, and to account for changes in the wind direction elsewhere than at the upstream turbines. However, the reviewer rightfully hints that this may not be the case in the situation with realistic, low-frequency changes in the inflow, as also mentioned in earlier comments. It is still uncertain what methodology works best for realistic inflow conditions.

  **Revision plan**: Since the wind direction estimation algorithm has not been tested under the relevant conditions, this algorithm will be presented to be rather a proposal or suggestion instead of an actual validated solution.

- **Feedback**: p.18, line 14: estimation -> you mean "parameter estimation"?

  **Response**: The sentence "*First, the performance of the ExKF, UKF, EnKF, and the case without estimation are compared for the two-turbine simulation case of Table 1.*" is expected to have caused confusion. Actually, in this section, four simulation cases are compared:

  - State estimation with Extended Kalman filter (ExKF)

- State estimation with Unscented Kalman filter (UKF)
- State estimation with Ensemble Kalman filter (EnKF)
- Open-loop, no estimation (OL)

None of the cases include parameter estimation. The authors will rephrase this sentence to make it clear.

**Revision plan**: This sentence will be rephrased for clarity.

- **Feedback**: p.21 Fig 6 caption: "The freestream wind is coming in from the top of the page, and flows towards the bottom." Isn't it the other way around?

  **Response**: The authors appreciate the reviewer's comment, and it now also becomes more apparent that this figure had not been presented clearly enough. The freestream wind flow is coming in from the top of the page and flowing towards the bottom, as correctly stated in the initial paper. The regions in red, i.e., the regions with a higher estimation error, are the waked regions, which are typically harder to predict than the freestream flow. This also corresponds to the way the measurements are defined: turbine power measurements are used in column 2, flow measurements *upstream* of each turbine are used in column 3, and flow measurements *downstream* of each turbine are used in column 4.

  **Revision plan**: An arrow indicating the freestream wind direction will be added to Figures 5, 6, and 9.

- **Feedback**: p.22 line 3: "Dual estimation using flow measurements downstream..." - please explain the used measurements in more detail

  **Response**: The authors appreciate the reviewer's detailed comments on the clarity of certain sections in the paper. Here, the downstream flow measurements refer back to the same downstream flow measurements presented in Section 4.2.2, as indicated in columns 2-4 in Figure 5 and column 4 of Figure 6.

  **Revision plan**: A clear reference to Section 4.2.2 and Figure 6 will be made on page 22, line 3.

- **Feedback**: p22, line 4: explain figure 7 clearly. What's on the y axis?

  **Response**: The variable on the $y$-axis refers back to the equation presented on page 19, line 2. Basically, this is the $L^2$ norm of the estimation error of all the longitudinal velocity states, $\boldsymbol{u}_k$, as defined in the equation on page 7, line 13. Note that Figures 5, 6, and 9 are graphical representations of the absolute value of the estimation errors of the same variable $\boldsymbol{u}_k$. These states are the most relevant for control, since the dominant wind direction is parallel to the $x$-axis (thus, aligned with $\boldsymbol{u}_k$).

  **Revision plan**: The authors will mention that the $y$-axis shows the $L^2$ norm, and explain it briefly. Furthermore, the authors will be more explicit concerning the flow fields presented in Figures 5, 6, and 9, stating that they show representations of the absolute value of the estimation errors of variable $\boldsymbol{u}_k$. This will additionally be linked back to Figure 7.

- **Feedback**: page 25, equation on bottom: bullet notation not clear

  **Response**: The variable $(\Delta P)^i$ represents the error between the true ("SOWFA") and the estimated ("OL" or "EnKF") turbine power signal timeseries of turbine $i$. Fundamentally, $(\Delta P)^i$ is the vector with errors between the forecasted and the true power signal over a certain time horizon. This forecast is from the current time instant, $T_f$, to the final time instant $T_k = 1000$ s. The $L^2$ norm of the timeseries of this variable for all turbines is shown in Figure 10. This could cause confusion, because it is not straight-forward how these values are derived from the $L^2$ norms for the single turbines. This is to be clarified in the revised document. Furthermore, the $\bullet$ notation is a placeholder for the respective method used for forecasting. This may be the open-loop forecast ("OL"), but can also be the forecast of the Ensemble Kalman filter ("EnKF").

  **Revision plan**: The authors will reformulate the equation presented on line 3 on page 25 in the revised article. It would be preferred to directly denote the definition of $||(\Delta P)_\bullet||_2$, rather than only the intermediate step to the equation. Furthermore, the bullet notation will be clarified in text.

**References**

S. Boersma, B M Doekemeijer, M Vali, J Meyers, and J W van Wingerden. A control-oriented dynamic wind farm model: WFSim. *Wind Energy Science*, pages 1–34, 2017. doi: 10.5194/wes-2017-44.

P M O Gebraad, F W Teeuwisse, J W van Wingerden, P A Fleming, S D Ruben, J R Marden, and L Y Pao. Wind plant power optimization through yaw control using a parametric model for wake effects - a CFD simulation study. *Wind Energy*, 19(1): 95–114, 2016. ISSN 1099-1824. doi: 10.1002/we.1822.

J P Goit and J Meyers. Optimal control of energy extraction in wind-farm boundary layers. *Journal of Fluid Mechanics*, 768:5–50, Apr 2015. doi: 10.1017/jfm.2015.70.

W Munters and J Meyers. An optimal control framework for dynamic induction control of wind farms and their interaction with the atmospheric boundary layer. *Philosophical Transactions of the Royal Society of London A: Mathematical, Physical and Engineering Sciences*, 375(2091), 2017. ISSN 1364-503X. doi: 10.1098/rsta.2016.0100.

J Sanz Rodrigo, R Aurelio Chávez Arroyo, P Moriarty, M Churchfield, B Kosović, P-E Réthoré, K S Hansen, A Hahmann, J D Mirocha, and D Rife. Mesoscale to microscale wind farm flow modeling and evaluation. *Wiley Interdisciplinary Reviews: Energy and Environment*, 6(2):e214, 2016. doi: 10.1002/wene.214.

C Santoni, E J García-Cartagena, U Ciri, G V Iungo, and S Leonardi. Coupling of mesoscale weather research and forecasting model to a high fidelity large eddy simulation. *Journal of Physics: Conference Series*, 1037(6), 2018.

---

## Author Response (AR1)

| | |
|---:|:---|
| Date | July 31, 2018 |
| Our reference | WES-2018-33 |
| Contact person | B M Doekemeijer |
| Telephone | +31 15 278 5623 |
| E-mail | b.m.doekemeijer@tudelft.nl |
| Subject | Response to reviewers |

**Delft University of Technology**

Delft Center for Systems and Control

Address
Mekelweg 2 (3ME building)
2628 CD Delft
The Netherlands

www.dcsc.tudelft.nl

Reviewers
*Wind Energy Science Discussions*

Dear Reviewers,

The authors express their gratitude to the reviewers for the time and effort spent to provide positive and constructive feedback to the submitted WES manuscript. Their comments have played a crucial role in further improving the scientific quality and relevance of this work. In accordance to the provided feedback, the article has been revised appropriately. The objective of the attached document is to respond to all the concerns raised by each reviewer, and explain how the authors have addressed each issue in the revised article.

Yours sincerely,

Bart Doekemeijer

Enclosure(s): Response to comments of Reviewer 1
Response to comments of Reviewer 2
Marked-up manuscript highlighting the changes made

Date     July 31, 2018
Reference  WES-2018-33

**Response to comments of Reviewer 1**

**General comments**

- Good and relevant work.
  We thank and appreciate the positive feedback of the reviewer.

**Specific comments**

- **Feedback**: I have concerns about the validity of neglecting the vertical dimension, given significant effects such as vertical meandering, wind shear and veer. There should be more comment about this.

  **Response**: The reviewer raises an important issue: the validity of the control-oriented model. This has also been a concern of the authors in previous work, and has been tackled in the original article of Boersma et al. [2017]. That paper presents and discusses the control-oriented model "WFSim" in greater detail, including the complete mathematical derivation and comparisons against two different high-fidelity large-eddy simulation codes. In the current WES submission, the aim is to provide a concise yet informative overview of the mathematical model used in the remainder of the article.

  Furthermore, from an estimation point of view, indeed the model should be as accurate as possible. However, from a controls perspective, the computational cost limits us to models that are fast enough for real-time application, and therefore the accuracy is limited. The presented estimation algorithm has the potential to account for modeling errors (e.g., vertical meandering, wind shear, and wind veer, as the reviewer rightfully mentioned) by assimilating real-time measurements into the simplified mathematical model. In addition to the simulation results presented in the work at hand, the simulation results presented in the work of Boersma et al. [2017] show that the model matches well in terms of the hub-height flow and the turbine power production compared to a high-fidelity large-eddy simulation code, which *does* include the aforementioned atmospheric effects. While these simulations are not conclusive, it at least provides a safe ground to state that the control-oriented model is valid for the demonstrated cases.

  **Revised changes**: The authors have clearly defined the scope of this paper, and more explicitly the scope of Section 2: "The scope of this section is ... assumptions made." Furthermore, the assumptions in the mathematical model are stated more explicitly, e.g., " Other vertical flow ... are neglected.". And the original article is referenced: "The reader is referred to Boersma et al. (2017b) for more information."

- **Feedback**: In the introduction, some re-wording is necessary: "all commercial turbines ... power electronics" is not right. They are certainly connected, it's only that the rotational speed is decoupled from the grid frequency. Also, in the next line, "grid-disconnected renewable energy plants": maybe "non-synchronous" or some other expression, but not "grid-disconnected"!

  **Response**: The reviewer is correct, and the proposed changes will be made to the introductory text.

  **Revised changes**: The text has been revised in accordance to the suggestions.
  - Page 1. *"While there are ... the rotational speed of almost all commercial turbines is currently decoupled from the electricity grid frequency via each turbine's power electronics (Aho et al., 2012). "*
  - Page 1. *"As the current grid-connected ... replaced by non-synchronous renewable energy plants, the inertia of the electricity grid will decrease."*

**Technical corrections**

- **Feedback**: Equation (5): Actually there is now plenty of evidence that the cos-cubed law for the effect of yaw on power coefficient is not at all correct. There should at least be some comment about this.

  **Response**: The authors thank the reviewer for raising this concern. This issue is in line with the comment concerning model validity. The reviewer is correct that other control-oriented models such as FLORIS [Gebraad et al., 2016] use a different correction term to incorporate a yaw misalignment in the turbine power expression. Unfortunately, to the best of the authors' knowledge, no systematic way is described in the literature on how to best choose this exponential term, besides through a comparison with high-fidelity data, as also discussed in the original paper by Boersma et al. [2017]. In the presented "WFSim" model, the cosine-cubed law originates from a physical derivation of the Navier-Stokes equations where the turbines are modeled using actuator disk theory. It might indeed turn out that, for a certain topology and turbine type, the cubed-law will not suffice, and a different exponent should be assumed. The effect of this discrepancy does not further impinge on the results presented in this article, as all simulations are with non-yawed turbines.

  **Revised changes**: In accordance with the comment of the author, the assumptions and limitations underlying the cosine-cubed law in Equation (5) are clearly mentioned, including an explicit reference to the original paper on WFSim by Boersma et al. [2017]. The original paper gives a more thorough motivation for this power law.

**Response to comments of Reviewer 2**

**General comments**

- **Feedback**: The paper is interesting, but way too long compared to its actual contribution. I suggest it is reconsidered for publication after major review.
  **Response**: We thank and appreciate the constructive feedback of the reviewer. The authors understand the concern of the reviewer, and have also had internal discussions before submission on whether or not to include certain sections. The purpose of this paper is to present a complete and unifying framework for joint state-parameter estimation for a control-oriented wind farm model, and therefore it was decided to include certain levels of detail. A more explicit motivation is found in the following responses.

**Major comments**

1. **Feedback**: My main comment is: reduce the length substantially to increase the readability, but most importantly, to clarify the actual contribution of the paper. Remove details that are previously published. More specifically,

   - in the introduction, focus on the actual innovation of the work and its relation to other work.
     **Response**: Currently, the complete closed-loop wind farm control framework is presented in Fig. 1, including the place of this work's contribution in relation to the other components. Furthermore, this (closed-loop) framework is compared against other (open-loop & model-free) frameworks on page 2. For each component of the presented framework, the state of the art is presented and compared to the work of the authors. The actual innovation of this work is described on page 3, where the current contributions compared to previous work are detailed.

- The detailed explanation of the surrogate model (Section 2) can be removed, providing the proper reference to the original work. Of course, a shorter summary of the model will be welcome.

  **Response**: The authors understand the concern of the reviewer. The authors had considered the removal of sections 2 and 3 before submission, but it was decided not to, because the purpose of this article is to present a unifying framework, including enough detail for unfamiliar readers (without the necessary aerodynamics and/or state estimation background) to understand the work to a sufficient degree. The model is presented in such detail to motivate the choice and understanding of the states and parameters that are to be estimated, and to help interpret the results found in section 4. For example, the mixing length parameter $\ell_s$ is estimated in certain simulations in section 4, and therefore the turbulence model is presented beforehand in section 2. Furthermore, the mathematical notation of section 2 is in agreement with that of sections 3 and 4, making it easier for unfamiliar readers to follow.

  **Revised changes**: The authors have revised the entire document and reduced the paper's length by $10$ %.

- Details about the different forms of (standard) Kalman filters are also not necessary, a reference is sufficient, like linear and extended KF.

  **Response**: We appreciate the reviewer's constructive feedback. This concern is in line with the previous comment concerning section 2. It is apparent that the reviewer is an expert in the field of control and state estimation. However, the target audience for this article is not exclusively control engineers, but to the wider range of the wind energy community. The purpose of this article is to present a unifying framework. The various Kalman filters are mathematically introduced to ensure reproducibility and to promote clarity and understanding when interpreting the results presented in section 4. The mathematical model formulation, the Kalman filtering section, and the results all maintain a homogeneous symbol notation and follow one another logically, providing clarity in the remainder of this article.

  **Revised changes**: The authors have revised the entire document and reduced the paper's length by $10$ %.

2. **Feedback**: Section 4 but wel written and clear, ...

   **Response**: We appreciate the positive feedback on section 4. The authors also believe that this clarity is largely due to the unifying framework presented, which led to an article that is longer than average.

   ...but I recommend the following changes:

Date    July 31, 2018
Reference    WES-2018-33

- **Feedback**: Sec 4.2.1 The improvement with respect to the estimation-free case does not seem to be very pronounced. Can you comment more on that. The state estimation adds significant computational complexity, which requires more significant benefits in terms of accuracy. Maybe a different simulation scenario, including changes in the wind conditions during the simulaton, can be used to better demonstrate the performance.

  **Response**: The reviewer addresses a very important topic: the usefulness of the estimator compared to the addition in computational cost. In Figure 5, an open-loop simulation ("WFSim") is compared to closed-loop simulations in which the states of WFSim are estimated using a variety of Kalman filters. For the 2-turbine simulation, the WFSim model is accurate for the first wake, and there is a negligible increase in accuracy. For the second wake, the improvements are more noticeable, and the presented simulation case suffices for the comparison of the different Kalman filter algorithms. The reviewer is correct to state that the case does not necessarily motivate the use of state estimation in the 2-turbine case. However, the 2-turbine simulation study showcases the differences in Kalman filters and sensor locations, building up towards the more realistic 9-turbine wind farm simulation. In the 9-turbine simulation study, the need for state estimation becomes more apparent.

  **Revised changes**: The purpose of the 2-turbine case study ("a two-turbine wind farm is simulated to analyze the effect of different measurement sources, KF algorithms, and the difference between state-only and state-parameter estimation") and the 9-turbine case study ("the purpose of this case study is to highlight the need for state-parameter estimation for accurate wind farm modeling") have been highlighted.

- **Feedback**: Sec 4.2.2, Fig 6: please show the model output (WT power) estimation and compare to the true output. At both turbines the wind speed is estimated clearly higher than the simulated one, implying that the estimated power will also be (much!) higher.

  **Response**: The authors appreciate the reviewer's idea of showing the turbine power signals, be it that it will elongate the article even further. However, the authors are confused about the statement that the estimated turbine power is higher than the simulated power. Figure 6 shows the absolute value of the *error* in the flow velocities. For clarification, the power signals are plotted next of 1) the simulation where the EnKF uses power measurements, 2) the simulation in which upstream flow measurements are used, and 3) the simulation in which downstream flow measurements are used:

[Figure]

**Power measurements**

[Figure]

**Upwind lidar measurements**

[Figure]

**Downwind lidar measurements**

From these figures, the difference between true and estimated power is small. Note that the red dots and red turbines in Fig. 6 of the paper are indicating that they use flow and turbine power measurements, respectively, rather than their error value. This may have caused confusion. The inflow at each turbine shows a close-to-blue color, indicating that the estimation error in wind speed is very small, which fits with the small error in power estimation from the figures above.

**Revised changes**: The authors have more carefully emphasized that what is being plotted in Figures 5, 6 and 9 are absolute values of the errors. This was done by additional information in the caption and main body text.

- **Feedback**: Fig 7: it is essential here to simulate using a realistic low frequency variations in the wind condition. Obviously, the prediction will be good when the incoming wind speed and direction does not change, since the underlying model does not change. The different KFs outperform the OL simulation in the prediction ONLY because the initial condition at the beginning of the prediction part is worse for the OL case (the KF based models have been adapted during the estimation phase, and the OL model has not). I suggest you focus in Figure 7 on the estimation part and remove the prediction part. Same holds for Figure 10.

**Response**: The authors thank the reviewer for the valuable comment and suggestions on further simulation studies. The authors think that it is not obvious that the predictions in Figure 7 will be accurate. Note that only two measurements are available at each timestep, while 1 model parameter and 3200 states are estimated simultaneously. As observability is not a given, the model is nonlinear and can go unstable, the turbines are following a quickly varying excitation signal, and we are feeding in noisy measurements of a significantly different, high-fidelity model, the authors believe it is not a given that the prediction will be accurate.

Furthermore, the purpose of showing the prediction is to highlight the difference in state-only against joint state-parameter estimation. In state-only estimation, indeed only the initial condition is better than the OL case. The state-only estimation forecast will converge to the OL forecast after a certain amount of time (around $t = 800$ s and $1100$ s, respectively). However, for the state-parameter estimation case, the estimate should outperform the state-only estimation for larger prediction horizons, showing the importance of parameter estimation in addition to state estimation.

Further, the authors think it is a great suggestion to investigate the performance under a realistic, time-varying and spatially-varying inflow. Currently, the turbulent inflow for SOWFA is generated following a precursor simulation, in which a realistic turbulent flow field is developed. As the reviewer rightfully mentions, while there are turbulent fluctuations in the inflow, this is with a single *mean* wind speed and wind direction. However, to generate a realistic inflow with low-frequency changes in the ambient conditions for a high-fidelity wind farm simulation, one would need to couple a mesoscale model with a large-eddy simulation model. This is a scientific study by itself (e.g., Rodrigo et al. [2016], Santoni et al. [2018]), and considered outside of the scope of this work.

The authors still believe there is value in the work presented in this article. Currently, the solution has been tested in a high-fidelity simulation environment with a realistic, turbulent inflow. Furthermore, convergence of the wind speed is shown, by purposely initializing the model internal to the estimation algorithm with a wrong freestream wind speed, with success. This suggests that in the comparable situation that when the actual freestream wind speed changes, the solution will succeed too.

An alternative to high-fidelity simulation would be to use experimental data. However, such data is currently difficult to obtain for the authors. Furthermore, it is questionable whether a wind tunnel provides an inflow with realistic low-frequency variations in ambient conditions, and flow scaling may become a problem too. Experimental full-scale data would be more sensible, but this will yield other issues such as confidentiality, the scientific relevance of the turbines in the farm, signal synchronization, sensor uncertainties and inconsistencies, and the lack of (reliable) flow and turbine measurements.

**Revised changes**: The need for high-fidelity data with realistic inflow conditions, including low-frequency changes in the wind direction, turbulence intensity, and wind speed has been presented as an important next step for future work, and the limitations of the current simulations have been described more explicitly. Important remarks and notes have been added at the start of the results section ("Also, note that the simulations... algorithm validation"), at the end of the results section ("A crucial remark... practical wind farm implementation"), and in the conclusion ("In future work... practical wind farm implementation").

- **Feedback**: Sec 4.3: the performance is compared to the open-loop simulation with the correct wind velocity U_inf, while the EnKF is initialized with a different U_inf. For better comparison I recommend you compare to the OL simulation with the same initial condition. That will not only be fair, but actually also in favor of your approach.

  **Response**: The reviewer has a good point: for a fair comparison, the authors should "compare apples to apples": use the same initial conditions in the OL simulation as in the KF simulation.

  **Revised changes**: The authors have added (rather than replaced) the OL simulation results with $U_\infty = 9$ m/s, including the necessary discussion and comparisons. Namely, the EnKF will easily outperform the OL simulation with $U_\infty = 9$ m/s even if it would only estimate the inflow conditions. It is interesting to also compare how well the EnKF performs compared to a pre-tuned OL simulation. The results are shown in Figures 9, 10, 11, and Table 5.

- **Feedback**: Fig 11. It would make more sense to focus the comparison only on the wind velocity in the wakes, rather than the whole wind field.

  **Response**: The reviewer makes a good point concerning the region of interest when comparing flow fields, and this has been a concern of the authors too in the past. However, the authors believe that it may also be of interest to accurately estimate the non-waked flow surrounding the wind turbines, as this may influence the optimal control strategy in the closed-loop framework later on. Namely, an optimal yaw steering strategy may depend on the non-waked flow conditions on either side of the farm. Furthermore, the definition of a waked region and the extraction of the flow velocities therein requires additional work and explanation, while the dominant trends are identical when considering the complete flow field. Note that the inclusion of the unwaked flow in the surrogate model is important for stability and to reduce boundary effects of the model.

3. **Feedback**: It is claimed on several occasions in the paper that the presented estimator is useful for performing long-term forcasting. I disagree with this statement, and if you want to convince me then you would have to demonstrate the ability of the estimator to make correct predictions in the future when the input conditions (eg ambient wind velocity) vary in time. For instance, to predict upcoming wind speed or direction changes. Obviously, this will not be possible, so I suggest that with respect to the application of the estimator you stick to feedback control. When used for MPC, I suggest using the term short-term prediction.

   **Response**: The reviewer makes a very good point, and there is not sufficient evidence to claim that our proposed solution can consistently and reliably provide long-term forecasting. Furthermore, after revision, the authors agree that the definition "long-term" has not been defined appropriately in the article. While the idea should not be discarded, there is not enough evidence supporting the claim as of right now. Further, the high-fidelity simulation with realistic low-frequency changes in the ambient conditions is outside of the scope of this article.

   **Revised changes**: As suggested, the claims concerning long-term forecasting have been removed. The claims for forecasting, including short-term forecasting, have been rephrased to address the limitations. Namely, time-varying wind directions and wind speeds have not been considered. In addition, the part on wind direction estimation in Section 3.7 has been removed, as it was neither tested in high-fidelity simulation nor with realistic low-frequency changes in the ambient conditions.

**Minor comments**

- **Feedback**: p.2, line 9 - remove bracket

  **Response**: The authors express gratitude for the time and effort invested by the reviewer to read the document so carefully.

  **Revised changes**: The suggested changes have been made.

- **Feedback**: p.6, line 2 - abbrev ADM seems not used in the sequel, check and if so - remove. Same holds for the abbrev. UT on page 11, line 15.

  **Revised changes**: The suggested changes have been made.

- **Feedback**: p.6, eq. 4: define $\phi$, D and $C'_{T_i}$ (what do you mean by variation?). Also, does the term H[] not imply that the thrust force is excerted within a circle around the turbine, rather than on the rotor plane/line?

  **Response**: With *variation*, a parametrization is meant. There is a one-to-one mapping between the traditional thrust coefficient and $C'_T$. The latter has been used more popularly in the work by Goit and Meyers [2015] and Munters and Meyers [2017], and this is the way it has been defined in the original paper by Boersma et al. [2017]. Furthermore, the reviewer is correct that $H[\bullet]$ implies a circle around the turbine center. However, the additional term $\delta[\bullet]$ projects this circle onto the rotor plane, leading to the actuator disk implementation.

  **Revised changes**: The symbols have been defined as suggested.

- **Feedback**: p. 8, bottom: notation already defined in Sec 2.4

  **Response**: The authors appreciate the reviewer's eye for detail.

  **Revised changes**: The repeated definition of symbols have been omitted.

- **Feedback**: p. 9 bottom: why is $P^z_{k|k-1}$ is not necessarily invertible, then $L_k$ will not be full rank (or possibly even $L_k = 0$), implying that some (or all) measurements will not contribute to the state estimation

  **Response**: The reviewer makes a justified comment concerning the invertibility of the covariance matrix, and the state updates that it results in. The issue of invertibility is closely related to how the covariance matrices are calculated. Specifically, for the sample-based algorithms, it may occur that singular covariance matrices arise (see Equation (28)). This is especially the case when the number of samples is smaller than the number of system states. As the reviewer rightfully mentions, this may result into a Kalman gain $L_k$ which is not full rank. In that situation, certain measurements may not be used to update the state vector. Since the samples in the Ensemble KF and thus the rank properties of the covariance matrix change in each timestep due to the random Gaussian noise, this is not expected to be an issue.

- **Feedback**: p.11, eq. (21): define all used notation

  **Revised changes**: The definition of $\overline{\psi}_{k-1|k-1}$ in Equation (21) has been introduced in the text, and the definition of $N$ has been repeated for clarity.

- **Feedback**: p.13, eq (27) : how are the estimates $\hat{\boldsymbol{w}}_{k-1}^i$ and $\hat{\boldsymbol{w}}_k^i$ obtained and updated?

  **Response**: The variables $\hat{\boldsymbol{w}}_{k-1}^i$ and $\hat{\boldsymbol{v}}_k^i$ are realizations of zero-mean Gaussian white noise, where the covariance is defined through Equation (8), which reads:

  $$\mathrm{E}\left[\begin{bmatrix} \boldsymbol{v}_k \\ \boldsymbol{w}_k \end{bmatrix} \begin{bmatrix} \boldsymbol{v}_\ell^T & \boldsymbol{w}_\ell^T \end{bmatrix}\right] = \begin{bmatrix} \boldsymbol{R}_k & \boldsymbol{S}_k^T \\ \boldsymbol{S}_k & \boldsymbol{Q}_k \end{bmatrix} \Delta_{k-\ell}, \quad \text{where} \quad \Delta_{k-\ell} = \begin{cases} 1, & \text{if } k = \ell, \\ 0, & \text{otherwise.} \end{cases}$$

  In practice, these noise terms are generated using MATLAB's `randn()` command, employing a constant preloaded random seed between simulations for one-to-one comparisons between different Kalman filtering algorithms.

  **Revised changes**: A more explicit explanation has been introduced near Equation (27), detailing how $\hat{\boldsymbol{w}}_{k-1}^i$ and $\hat{\boldsymbol{v}}_k^i$ are calculated.

- **Feedback**: p 16, line 1: are you suggesting to use the wind vane measurements of only the upstream turbines to estimate $\phi$, or all turbines? Please be clear, because if you use all turbines you will be neglecting the dynamics of the propagation of the wind direction through the wind farm.

  **Response**: The reviewer raises an important question concerning the determination of the wind direction, $\phi$. Currently, the wind direction is calculated as the average of the wind vane measurements of *all* turbines inside the farm, both up- and downstream. The issue that the reviewer mentions has not explicitly been considered in this work. Based on the available data from high-fidelity large-eddy simulations, no significant differences were found between the wind vane measurements for the various wind turbines. The main motivation for the use of all the turbine vane measurements was to reduce the variance by using all measurements as they are assumed to measure the same thing with (more or less) independent errors, and to account for changes in the wind direction elsewhere than at the upstream turbines. However, the reviewer rightfully hints that this may not be the case in the situation with realistic, low-frequency changes in the inflow, as also mentioned in earlier comments. It is still uncertain what methodology works best for realistic inflow conditions.

  **Revised changes**: Since the wind direction estimation algorithm has not been tested under the relevant conditions, this wind direction estimation algorithm has been removed from the revised article.

- **Feedback**: p.18, line 14: estimation -¿ you mean "parameter estimation"?

  **Response**: The sentence *"First, the performance of the ExKF, UKF, EnKF, and the case without estimation are compared for the two-turbine simulation case of Table 1."* is expected to have caused confusion. Actually, in this section, four simulation cases are compared:

  - State estimation with Extended Kalman filter (ExKF)
  - State estimation with Unscented Kalman filter (UKF)
  - State estimation with Ensemble Kalman filter (EnKF)
  - Open-loop, no estimation (OL)

  None of the cases include parameter estimation. The authors will rephrase this sentence to make it clear.

  **Revised changes**: This sentence has been rephrased for clarity.

- **Feedback**: p.21 Fig 6 caption: "The freestream wind is coming in from the top of the page, and flows towards the bottom." Isn't it the other way around?

  **Response**: The authors appreciate the reviewer's comment, and it now also becomes more apparent that this figure had not been presented clearly enough. The freestream wind flow is coming in from the top of the page and flowing towards the bottom, as correctly stated in the initial paper. The regions in red, i.e., the regions with a higher estimation error, are the waked regions, which are typically harder to predict than the freestream flow. This also corresponds to the way the measurements are defined: turbine power measurements are used in column 2, flow measurements *upstream* of each turbine are used in column 3, and flow measurements *downstream* of each turbine are used in column 4.

  **Revised changes**: There appear to be some artifacts with the images, in which the orientation of the flow fields depend on the document viewer used. The authors have attempted to repair this. In the author's document viewers, the orientation is as described in the caption.

- **Feedback**: p.22 line 3: "Dual estimation using flow measurements downstream..." - please explain the used measurements in more detail

  **Response**: The authors appreciate the reviewer's detailed comments on the clarity of certain sections in the paper. Here, the downstream flow measurements refer back to the same downstream flow measurements presented in Section 4.2.2, as indicated in columns 2-4 in Figure 5 and column 4 of Figure 6.

  **Revised changes**: A clear reference to Section 4.2.2 and Figure 6 has been made on page 22, line 3.

Date  July 31, 2018

Reference  WES-2018-33

- **Feedback**: p22, line 4: explain figure 7 clearly. What's on the y axis?

  **Response**: The variable on the $y$-axis refers back to the equation presented on page 19, line 2. Basically, this is the $L^2$ norm of the estimation error of all the longitudinal velocity states, $\boldsymbol{u}_k$, as defined in the equation on page 7, line 13. Note that Figures 5, 6, and 9 are graphical representations of the absolute value of the estimation errors of the same variable $\boldsymbol{u}_k$. These states are the most relevant for control, since the dominant wind direction is parallel to the $x$-axis (thus, aligned with $\boldsymbol{u}_k$).

  **Revised changes**: The authors mention that the $y$-axis shows the $2-$norm. Furthermore, the authors have rephrased the captions of Figures 5, 6, and 9, stating that they show representations of the absolute value of the estimation errors of variable $\boldsymbol{u}_k$.

- **Feedback**: page 25, equation on bottom: bullet notation not clear

  **Response**: The variable $(\Delta P)^i$ represents the error between the true ("SOWFA") and the estimated ("OL" or "EnKF") turbine power signal timeseries of turbine $i$. Fundamentally, $(\Delta P)^i$ is the vector with errors between the forecasted and the true power signal over a certain time horizon. This forecast is from the current time instant, $T_f$, to the final time instant $T_k = 1000$ s. The $L^2$ norm of the timeseries of this variable for all turbines is shown in Figure 10. This could cause confusion, because it is not straight-forward how these values are derived from the $L^2$ norms for the single turbines. This is to be clarified in the revised document. Furthermore, the $\bullet$ notation is a placeholder for the respective method used for forecasting. This may be the open-loop forecast ("OL"), but can also be the forecast of the Ensemble Kalman filter ("EnKF").

  **Revised changes**: The authors have decided to remove the definition of $||(\Delta P)_\bullet||_2$, and rather put all the results in a separate table (Table 5), with a more insightful description and definition of the variable represented.

Date    July 31, 2018
Reference    WES-2018-33

    (i) Each particle is propagated forward in time using the nonlinear system dynamics, and with the realizations of noise terms $w$ and $v$ denoted by $\hat{w}^i_{k-1} \in \mathbb{R}^N$ and $\hat{v}^i_k \in \mathbb{R}^M$,  generated using MATLABs *randn*(...) function.

$$\psi^i_{k|k-1} = f(\psi^i_{k-1|k-1}, q_{k-1}) + \hat{w}^i_{k-1} \qquad \text{for} \quad i = 1,...,Y,$$
$$\zeta^i_{k|k-1} = h(\psi^i_{k|k-1}, q_k) + \hat{v}^i_k \qquad \text{for} \quad i = 1,...,Y. \
[revised manuscript text omitted]

---

## Author Response (AR2)

Dear reviewer,

We appreciate your valuable feedback. This further improves the scientific quality and relevance of the article. Find attached our responses, and a manuscript showing the changes we made accordingly.

Best regards,
Bart Doekemeijer

- *RC1: Thanks for your detailed answers to my points.*
  *I still think the paper is good in general, but way too long for its actual contribution.*
  Thank you for your positive assessment. There is always a trade-off between clarity of the presentation and the length of a paper.

- *RC2: I suggested some parts to be removed that are known or available in other publications, and to just refer to them. Your point that the audience should not necessarily know what a KF is valid, but not convincing to retain the KF definitions (the audience would anyway not learn KF from your paper). The model also is presented elsewhere.*
  Dear reviewer. We appreciate your feedback on the paper, and we shortened the paper. The chapter on the surrogate model has significantly been shortened. Furthermore, the part on linearized Kalman filtering and the equations of the Extended Kalman filter have been omitted in the revised version. This further reduces the paper by 3 pages, in addition to the 3-page reduction in the last revision. We sincerely hope that you will consider this a sufficient reduction.

- *RC3: The open loop prediction by the KFs is, as far as I am concerned, also of little value. With or without parameter estimation included, mathematically, I don't see why an (extended) KF should necessarily perform better than an OL model when there are no measurements (provided that the OL model is not unstable, which should be the case). In the paper you state that in Fig 7 for predictions after the dashed line no measurements are used by the models, while in your response to my comment you write "...we are feeding in noisy measurements..." as argument for the better performance, which is rather embarrassing?!? I would recommend to completely remove Fig 7, unless the prediction is made under different wind conditions than during the closed-loop operation.*
  Dear reviewer. We apologize for causing any possible confusion. For the simulations with joint state-parameter estimation, the open-loop (OL) model and the Kalman filters are started with the same initial value for $l_s$. Then, every second, (noisy) measurements are fed into the KFs, and the state vector as well as the model parameter $l_s$ are estimated.  However, for the open-loop simulation, no measurements are fed in: the state vector is simply updated with the nominal model, and the value for $l_s$  remains the same.
  Now, after 600 s (left plot in Fig. 6) and 900 s (right plot in Fig. 6), we start a forecast, meaning no measurements are available after this time. At this moment, the OL model still has the same (poor) value for $l_s$  as at t=0 s, but the value for $l_s$  in the EnKF is different (better, hopefully). This leads to the OL and joint KF forecasts to differ from each other. Noticeably, in terms of control, $l_s$  is assumed to be a random walk model when estimated with the EnKF, essentially putting a pole at zero (cont. time), making the system marginally stable. In that case, the OL and KF solutions do not necessarily converge anymore. This is indeed what we see in Fig. 6. For state-only estimation, we have a stable system, and the forecasts converge within 100-200 seconds. For joint state-parameter estimation, we have a marginally stable system, and the forecasts do not converge. Actually, since the EnKF-joint and UKF-joint algorithms do their forecast with more accurate values for $l_s$, their forecasts have lower errors.

  For clarification, I have added a more elaborate explanation of the effects of parameter estimation in Section 4.2.3.

[revised manuscript text omitted]